# Ribosomal protein S27-like is a physiological regulator of p53 that suppresses genomic instability and tumorigenesis

Xiufang Xiong[1†], Yongchao Zhao[1†], Fei Tang[2], Dongping Wei[1], Daffyd Thomas[3,4], Xiang Wang[1], Yang Liu[2], Pan Zheng[2], Yi Sun[1,5]*

[1]Division of Radiation and Cancer Biology, Department of Radiation Oncology, University of Michigan, Ann Arbor, United States; [2]Center for Cancer and Immunology Research, Children's National Medical Center, Washington, United States; [3]Department of Pathology, University of Michigan Medical School, Ann Arbor, United States; [4]Comprehensive Cancer Center, University of Michigan Medical School, Ann Arbor, United States; [5]Institute of Translational Medicine, Zhejiang University School of Medicine, Hangzhou, China

**Abstract** Cell-based studies showed that several Mdm2-binding ribosomal proteins, upon overexpression, stabilize and activate p53. In contrast, here we show in a mouse knockout study that Mdm2-binding ribosomal protein S27-like (Rps27l), upon disruption, activates p53. Germline inactivation of *Rps27l* triggers ribosomal stress to stabilize Mdm2, which degrades Mdm4 to reduce Mdm2-Mdm4 E3 ligase towards p53, leading to p53-dependent apoptotic depletion of hematopoietic stem cells and postnatal death, which is rescued by *Trp53* deletion. Paradoxically, while increased p53 is expected to inhibit tumorigenesis, *Rps27l*$^{-/-}$;*Trp53*$^{+/-}$ mice develop lymphomas at higher incidence with p53 loss-of-heterozygosity and severe genome aneuploidy, suggesting that *Rps27l* disruption impose a selection pressure against p53. Thus, Rps27l has dual functions in p53 regulation: under *Trp53*$^{+/+}$ background, *Rps27l* disruption triggers ribosomal stress to induce p53 and apoptosis, whereas under *Trp53*$^{+/-}$ background, *Rps27l* disruption triggers genomic instability and *Trp53* deletion to promote tumorigenesis. Our study provides a new paradigm of p53 regulation.

*For correspondence: sunyi@ umich.edu

†These authors contributed equally to this work

## Introduction

Tumor suppressor p53 is a key regulator of cell growth and cell death (*Ko and Prives, 1996*; *Levine, 1997*) and is activated by many environmental stimuli, including DNA damaging agents (*Giaccia and Kastan, 1998*). Activated p53 acts as a guardian of the genome by inducing growth arrest, allowing cells to repair the damage, or apoptosis when the damage is too severe and irreparable (*Vogelstein et al., 2000*; *Levine and Oren, 2009*). p53 is frequently inactivated during human carcinogenesis either by point mutation which occurs in 50% of human cancers (*Greenblatt et al., 1994*), or by Mdm2/ Mdm4-mediated ubiquitylation and degradation (*Kruse and Gu, 2009*; *Wade et al., 2013*).

Mdm2 is a direct p53 target. Upon induction by p53, Mdm2 inactivates p53 through at least two main mechanisms: (a) binding to p53 at its transactivation domain and blocking its transactivation activity, and (b) serving as an E3 ubiquitin ligase to promote a rapid degradation of p53 (*Haupt et al., 1997*; *Honda et al., 1997*; *Kubbutat et al., 1997*; *Fang et al., 2000*; *Honda and Yasuda, 2000*). Thus, p53-Mdm2 forms an auto-regulatory feedback loop to keep p53 levels under control (*Wu et al., 1993*). Both in vitro and in vivo studies indicated that oncogenic activity of Mdm2 is mainly attributable to its binding to and degrading p53 (*Jones et al., 1995*; *Montes de Oca Luna et al., 1995*;

**eLife digest** There are over a hundred different types of cancer that can affect humans; but, in general, all cancers are caused by mutations that cause cells to grow and divide abnormally. 'Tumor suppressor genes' are genes that normally protect a cell from genetic changes that can lead a cell towards becoming cancerous.

About half of all cancers in humans have a mutation in one of the two copies of a tumor suppressor gene that encodes a protein called p53, which helps to control how and when cells grow and divide. In normal cells, the p53 protein can be activated in various ways. Damage to a cell's DNA triggers p53 to stop the cell growing, which gives the cell time to repair the DNA damage. However, if the damage is too severe and cannot be repaired, p53 essentially causes the cell to kill itself, via a process called apoptosis. Furthermore, if a cell has problems building new copies of its protein-making machinery, some of the parts (called ribosomal proteins) that make up these molecular machines can also lead to p53 being activated.

By deleting the gene for a protein called Rps27l that is a newly characterized ribosomal protein, Xiong et al. have discovered that, in mice, Rps27l regulates the p53 protein in two different ways. In normal cells, Rps27l appears to inhibit p53, which is likely to encourage cancer to develop. But, if a cell has already lost a copy of the p53 gene—a situation that would normally encourage the cells to accrue further mutations and become cancerous—Rps27l acts as a tumor suppressor. In these mutated cells, the Rps27l protein helps to maintain the stability of the genome and prevent the loss of the second copy of gene for p53, and so protects the cell from becoming cancerous.

Thus Rps27l can either activate or inactivate p53 activity depending on how many copies of the gene for p53 remain intact. The next challenge is to investigate if Rps27l levels determine the early-onset of tumor development in cancer-prone cells seen in patients with Li-Fraumeni syndrome, who are born with a mutated copy of the p53 gene.

*de Rozieres et al., 2000*). The key role of Mdm2 E3 ligase activity in controlling p53 levels in a physiological setting was further demonstrated by a knock-in study in which introduction of a ligase dead *Mdm2^{C462A}* mutant results in embryonic lethality, that like *Mdm2* deletion, can be fully rescued by simultaneous *Trp53* deletion (*Itahana et al., 2007*).

In addition to Mdm2, p53 is also subject to negative regulation by Mdm4 (also known as MdmX), an Mdm2 family member (*Shvarts et al., 1996*). Although Mdm4 itself does not have an intrinsic E3 ligase activity toward p53 (*Linares et al., 2003*), it does bind to p53 transactivation domain to block its transcription activity (*Shvarts et al., 1996*). Moreover, Mdm4 forms a tight 1:1 complex with Mdm2 via their respective C-terminal RING finger domains (*Sharp et al., 1999*; *Tanimura et al., 1999*), and the Mdm2-Mdm4 heterodimers are the preferred dimer form, compared to the Mdm2-Mdm2 or Mdm4-Mdm4 homodimers (*Kostic et al., 2006*). Furthermore, Mdm4 is a direct substrate of Mdm2 for targeted ubiquitylation and degradation (*de Graaf et al., 2003*; *Kawai et al., 2003*; *Pan and Chen, 2003*). More importantly, both in vitro cell culture studies using Mdm2 mutants (*Kawai et al., 2007*; *Poyurovsky et al., 2007*; *Uldrijan et al., 2007*) and in vivo studies using knock-in mice of Mdm2 and Mdm4 mutants (*Itahana et al., 2007*; *Huang et al., 2011*; *Pant et al., 2011*; *Wang et al., 2011*) demonstrated that the Mdm2-Mdm4 heterodimer has an optimal E3 ligase activity and is required for p53 degradation. Thus, the Mdm2-Mdm4 complex is interconnected and cross-regulated to keep p53 levels precisely in check under physiological conditions (such as during embryogenesis) and in response to various stresses (*Wade et al., 2013*).

Accumulated biochemical and cellular studies have shown that the p53-MDM2-MDM4 axis is further regulated by various ribosomal proteins (*Zhang and Lu, 2009*). Specifically, the ribosomal proteins, such as RPL11 (*Lohrum et al., 2003*; *Zhang et al., 2003*; *Bhat et al., 2004*; *Sasaki et al., 2011*), RPL5 (*Dai and Lu, 2004*), RPL23 (*Dai et al., 2004*; *Jin et al., 2004*), RPL26 (*Zhang et al., 2010*), RPS7 (*Chen et al., 2007*; *Zhu et al., 2009*), RPS3 (*Yadavilli et al., 2009*), RPS27/S27L (*Xiong et al., 2011*), S27a (*Sun et al., 2011*), RPS25 (*Zhang et al., 2013*), RPS26 (*Cui et al., 2014*) and RPS14 (*Zhou et al., 2013*), as well as RPL37, RPS15 and RPS20 (*Daftuar et al., 2013*), were found to bind to MDM2 upon ribosomal stress and inhibit MDM2-mediated p53 ubiquitylation and degradation, leading to p53 activation to induce growth arrest and apoptosis, thus acting as p53 activators (*Zhang and Lu, 2009*).

However, whether and how these Mdm2-binding ribosomal proteins indeed regulate p53 by modulating Mdm2 activity has not been explored previously using an in vivo mouse model.

RPS27L (NM_015920) is an 84-amino acid ribosomal like protein, which differs from its family member RPS27 (NM_001030) by only three amino acids (R5K, L12P, K17R) at the N-terminus. We and the others found that RPS27L is a direct p53 inducible target (*He and Sun, 2007*; *Li et al., 2007*). Our recent cell-based study showed that RPS27L directly binds to MDM2 and is subjected to MDM2-mediated ubiquitylation and degradation (*Xiong et al., 2011*). Furthermore, RPS27L competes with p53 for MDM2 binding, consequently inhibiting MDM2-mediated p53 ubiquitylation and degradation (*Xiong et al., 2011*). Thus, RPS27L interplays with the MDM2-p53 axis to regulate p53 activity. Although several ribosomal proteins have been previously shown to bind and inhibit MDM2, causing p53 activation (*Zhang and Lu, 2009*), with RPS7 and RPL26 being MDM2 substrates as well (*Ofir-Rosenfeld et al., 2008*; *Zhu et al., 2009*), RPS27L is the first and only known ribosomal-like protein that is a direct p53 inducible target, a MDM2 substrate, and a regulator of the MDM2-p53 axis. However, the physiological function of Rps27l and whether Rps27l plays a physiological role in regulation of the p53-Mdm2-Mdm4 axis in mouse remain entirely unknown.

Here we present in vivo evidence that, unlike in vitro cell culture studies which showed that several Mdm2-binding ribosomal proteins act as p53 activators, Rps27l, under the *Trp53^{+/+}* background, appears to be a physiological p53 inhibitor that stabilizes the Mdm2-Mdm4 heterodimer for effective p53 ubiquitylation and degradation. Unexpectedly, we also found that Rps27l, under the *Trp53^{+/−}* background, acts as a tumor suppressor by maintaining the genomic integrity to prevent the loss of *Trp53* heterozygosity and subsequent development of lymphoma. Thus, Rps27l regulates p53 either negatively or positively in a manner dependent of *Trp53* dosage.

## Results

### *Rps27l* disruption causes postnatal death as a result of increased apoptosis

Our previous studies showed that RPS27L is a direct p53 target (*He and Sun, 2007*) and regulates p53 activity by interacting with MDM2 (*Xiong et al., 2011*). To test the physiological function of Rps27l, we generated the gene-trap based *Rps27l^{+/−}* mice through Texas Institute for Genomic Medicine (TIGM) in a pure C57BL/6 background from an ES clone, IST11658B7, with a targeted BGEO vector inserted at the intron 1 to disrupt the open reading frame of *Rps27l* (*Figure 1—figure supplement 1A*). Intercrossing of *Rps27l^{+/−}* mice gave rise to the F1 offspring with three *Rps27l* genotypes, as confirmed by Southern blotting, PCR genotyping, and immunoblotting (*Figure 1—figure supplement 1B–D*). Thus, *Rps27l* deletion is not embryonic lethal. However, genotyping of 351 offspring at weaning of the *Rps27l^{+/−}* intercrossing failed to identify any *Rps27l^{−/−}* mice (*Figure 1A*). The ratio of *Rps27l^{+/+}* vs *Rps27l^{+/−}* mice was about 1:2, indicating that *Rps27l* mutation is homozygous lethal (*Figure 1A*). In fact, all *Rps27l^{−/−}* pups in a pure C57BL/6 background died within 12 days of birth. Under a mixed sv129/B6 background, about 50% mice die at 18 day after birth with the longest life-span of 35 days (*Figure 1—figure supplement 1E*). Thus, *Rps27l* disruption causes postnatal death.

Compared to the wild type and heterozygous littermates, *Rps27l^{−/−}* mice were about 50% in size and body weight (*Figure 1B,C*). At the organ levels after normalization with the body weight, the weight in the thymus and spleen was significantly reduced in *Rps27l^{−/−}* mice at P8–10 (*Figure 1D*). The H&E staining revealed that the bone marrow derived from *Rps27l^{−/−}* mice have much reduced cellularity (*Figure 1E*). Likewise, the *Rps27l^{−/−}* thymus showed a remarkable reduction in the thymic cortex, and thymocytes in cortical and cortical-medullary junction areas undergo apoptosis with accumulation of small irregular nuclear fragments (*Figure 1F*, arrows). Induction of apoptosis was further confirmed by the immunofluorescent staining with antibody against cleaved caspase-3 in both thymocytes (*Figure 1G*) and bone marrow cells (*Figure 1H*). Moreover, apoptosis in thymus was readily detectable by immunoblotting, which showed increased levels of cleaved caspase-3 and pro-apoptotic protein Puma (*Figure 1I*). An increased apoptosis, as reflected by Annexin V staining (*Figure 1J*, *Figure 1—figure supplement 1F*), but not reduced proliferation, as reflected by BrdU incorporation (*Figure 1—figure supplement 1G*) was also seen in myeloid progenitors (MP), derived from E14.5 *Rps27l^{−/−}* fetal livers. Finally, hypo-cellularity of bone marrow, thymus, and spleen was readily observed in *Rps27l^{−/−}* mice with a mixed 129/B6 background (*Figure 1—figure supplement 1H,I*). Thus, Rps27l is required for postnatal development of some organs, particularly thymus, spleen and bone marrow,

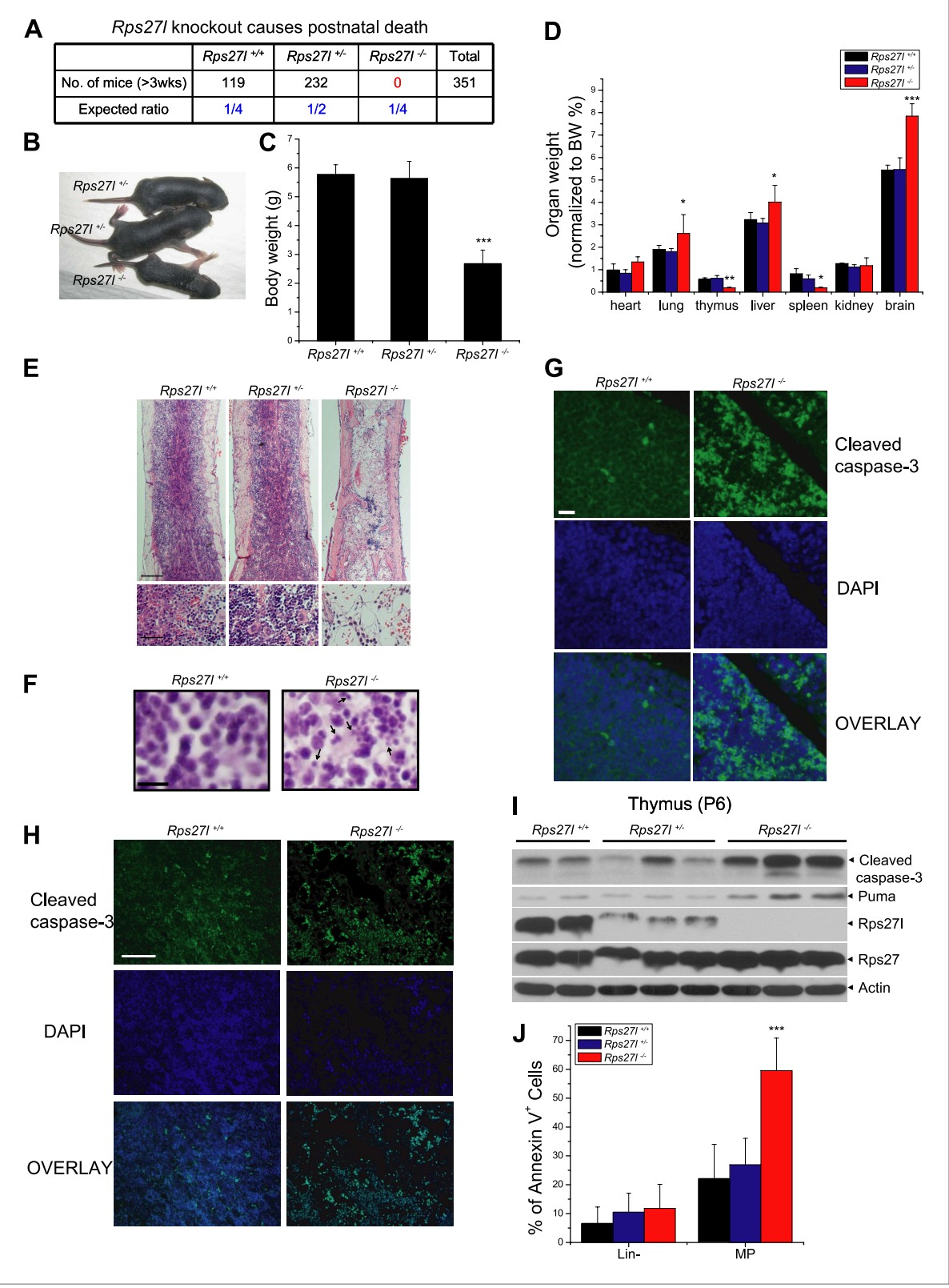

**Figure 1**. *Rps27l* disruption causes postnatal death as a result of increased apoptosis. (**A**) Disruption of *Rps27l* causes postnatal death. (**B–D**) Reduced body size, body weight, and organ weight in *Rps27l⁻/⁻* mice. An *Rps27l⁻/⁻* pup (bottom) and two *Rps27l⁺/⁻* littermates (top) at P8 were photographed (**B**). The body (**C**) and the organs (**D**) of P8–10 pups with genotypes of *Rps27l⁺/⁺* (n = 3), *Rps27l⁺/⁻* (n = 6), and *Rps27l⁻/⁻* (n = 10) were weighed. Shown are *Figure 1. Continued on next page*

Figure 1. Continued

mean ± SD. *p < 0.05, **p < 0.01, and ***p < 0.001, as compared to $Rps27l^{+/+}$ counterparts. (**E**) Representative H&E staining of bone marrows in femurs of P6 pups. Scale bars represent 200 μm (top) and 40 μm (bottom). (**F**) Representative H&E staining of thymuses from P8–10 pups. Arrows point to apoptotic cells. Scale bar represents 10 μm. (**G** and **H**) Representative immunofluoresent staining of cleaved caspase-3 in thymuses (**G**) of P8–10 pups and bone marrows (**H**) of P6 pups. Scale bars represent 20 μm (**G**) and 100 μm (**H**). (**I**) Accumulation of cleaved caspase-3 and Puma in $Rps27l^{-/-}$ thymuses. The thymuses of P6 pups were lysed for immunoblotting (IB). (**J**) Increased Annexin V-positive myeloid progenitors (MP) in $Rps27l^{-/-}$ fetal livers. Cells from E14.5 $Rps27l^{+/+}$ (n = 6), $Rps27l^{+/-}$ (n = 12), and $Rps27l^{-/-}$ (n = 11) fetal livers were stained with antibodies (Abs) against surface markers and Annexin V-FITC, followed by FACS analysis. Data shown are mean ± SD. ***p < 0.001, as compared to $Rps27l^{+/+}$ counterparts.

The following figure supplement is available for figure 1:

**Figure supplement 1**. Generation of $Rps27l$ gene trap mice and phenotypes of $Rps27l^{-/-}$ mice.

and postnatal death of $Rps27l$ knockout mice is likely associated with enhanced apoptosis, leading to bone marrow depletion.

## $Rps27l$ disruption causes the loss of hematopoietic stem and progenitor cells in fetal liver

To determine potential sources of bone marrow depletion in $Rps27l^{-/-}$ pups, we compared the fetal livers from E14.5 embryos among three $Rps27l$ genotypes and found that $Rps27l^{-/-}$ fetal livers are visibly smaller with significant reduction in cell number (**Figure 2A,B**). The FACS analysis of $Rps27l^{-/-}$ fetal livers revealed a substantial reduction in number of hematopoietic stem and progenitor cells (HSPCs), including hematopoietic stem cells (HSCs)-containing LSK (Lin⁻/Sca-1⁺/c-Kit⁺) population, and MP population, consisting of common myeloid progenitor (CMP), granulocyte-monocyte progenitor (GMP), and megakaryocytic-erythroid progenitor (MEP) cells (**Figure 2C–F**, **Figure 2—figure supplement 1A**). To determine the viability and functionality of $Rps27l^{-/-}$ HSPCs, we performed a non-competitive reconstitution assay to see whether these stem cells, while reduced in number, are still sufficient to rescue the bone marrow failure of recipient mice which were sterilized by a lethal dose of radiation. A total of $2 \times 10^6$ fetal liver cells from either $Rps27l^{+/+}$ or $Rps27l^{-/-}$ embryos were used. Remarkably, while HSPCs from wild type fetal liver cells completely reconstituted sterilized bone marrow and fully rescued the recipient mice, cells from $Rps27l^{-/-}$ fetal livers were unable to do so, leading to a 100% death of recipient mice within 10 days of reconstitution (**Figure 2G,H**, **Figure 2—figure supplement 1B**). To eliminate possibility that the failure in rescue is due to an insufficient number of stem cells in $Rps27l^{-/-}$ fetal livers, which is about 50% of the $Rps27l^{+/+}$ control (**Figure 2C**, 0.005% vs 0.01% of total liver), we repeated this non-competitive reconstitution assay, using three times more fetal liver cells ($6 \times 10^6$) from the $Rps27l^{-/-}$ embryos than that ($2 \times 10^6$) from the $Rps27l^{+/+}$ or $Rps27l^{+/-}$ embryos. Again, the stem cells from $Rps27l^{-/-}$ fetal livers failed to rescue sterilized recipient mice with 100% death within 12 days post reconstitution, whereas those from wild type or heterozygous fetal liver caused a 100% rescue (**Figure 2—figure supplement 1C**), indicating an intrinsic defect. Peripheral blood profiling of rescued mice showed a normal distribution of various blood cells (**Figure 2—figure supplement 1D,E**).

To further confirm the inability of stem/progenitor cells from $Rps27l^{-/-}$ fetal livers in re-building the recipient bone marrow, we performed a competitive reconstitution assay in which a 1:4 mixture of recipient bone marrow cells with donor fetal liver cells derived from $Rps27l^{+/+}$ vs $Rps27l^{-/-}$ embryos was given to sterilized recipient mice (**Figure 2I**). In this case, all recipient mice survived, as expected. Bone marrow profiling of surviving chimeric mice at 4, 12 and 20 weeks post reconstitution revealed that while 65–85% cells were derived from the $Rps27l^{+/+}$ donor fetal livers, none of cells were derived from $Rps27l^{-/-}$ donor fetal livers (**Figure 2J,K**). Together, our study clearly demonstrated in vivo that HSPCs from $Rps27l^{-/-}$ fetal livers must gradually die or fail to repopulate sterilized bone marrow. Thus, defective hematopoiesis is most likely contributable to postnatal lethality of $Rps27l^{-/-}$ mice.

## $Rps27l$ disruption increases p53 level

We next investigated potential mechanisms by which $Rps27l$ disruption induces apoptosis and causes the loss of HSPCs and depletion of bone marrow. We focused on p53, since (1) our recent study showed that RPS27L could modulate MDM2 to regulate the level and activity of p53 (**Xiong et al., 2011**), and (2) growth retardation in $Rps27l^{-/-}$ mice and apoptosis in $Rps27l^{-/-}$ organs (e.g., thymus and

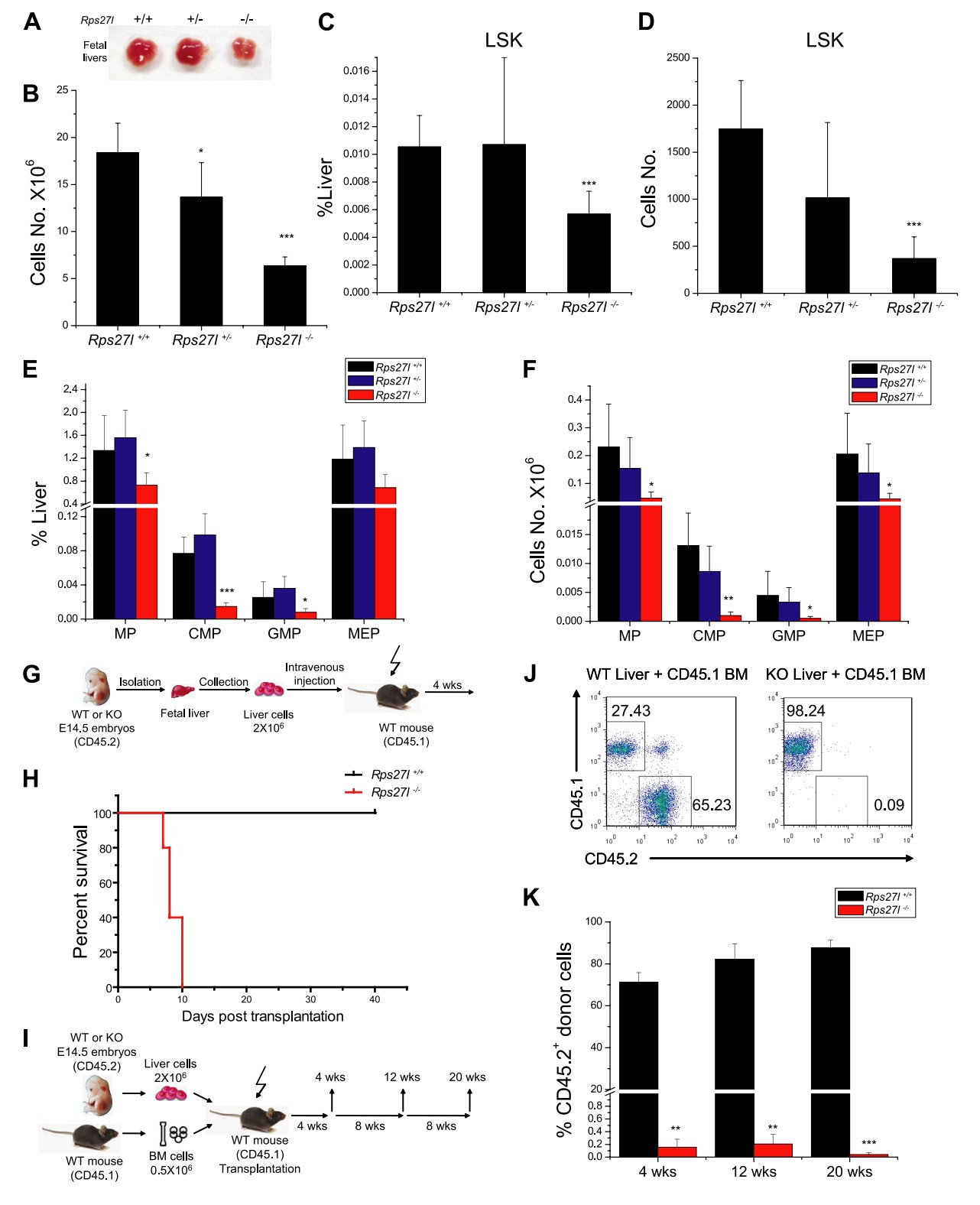

**Figure 2**. Reduced hematopoietic stem and progenitor cells in *Rps27l⁻/⁻* fetal livers. (**A** and **B**) Reduced size and total cell numbers in *Rps27l⁻/⁻* fetal livers. Representative fetal livers of three genotypes were photographed (**A**), and the numbers of fetal liver cells from E14.5 embryos were counted. Shown are mean ± SD with embryo numbers as follows: *Rps27l⁺/⁺* (n = 7), *Rps27l⁺/⁻* (n = 10), and *Rps27l⁻/⁻* (n = 8). *p < 0.05, ***p < 0.001, as compared with *Rps27l⁺/⁺* counterparts. (**C–F**) Decreased percentages and absolute numbers of HSPCs in *Rps27l⁻/⁻* fetal livers. Cells from E14.5 fetal livers of *Figure 2. Continued on next page*

*Figure 2. Continued*

$Rps27l^{+/+}$ (n = 7), $Rps27l^{+/-}$ (n = 10), and $Rps27l^{-/-}$ (n = 8) were stained with antibodies against various surface markers. The populations of LSK (**C** and **D**), MP, CMP, GMP, and MEP (**E** and **F**) were analyzed by flow cytometry. Shown are mean ± SD. *p < 0.05, **p < 0.01, and ***p < 0.001, as compared with $Rps27l^{+/+}$ counterparts. (**G**) Diagram of non-competitive reconstitution assay. Fetal liver cells (2 × 10⁶ cells) from E14.5 embryos (CD45.2) were injected into lethally irradiated recipient mice (CD45.1). Peripheral blood from recipients was analyzed by flow cytometry at 4 weeks after transplantation. (**H**) Kaplan–Meier survival curves of recipient mice after transplantation. $Rps27l^{+/+}$ or $Rps27l^{-/-}$ fetal liver cells were injected, respectively, into recipient mice (n = 5, for each genotype). p = 0.0026. (**I**) Diagram of competitive reconstitution assay. Fetal liver cells (2 × 10⁶ cells) from E14.5 embryos (CD45.2) were injected into lethally irradiated recipient mice (CD45.1) together with bone marrow cells (0.5 × 10⁶ cells) from normal recipient mice. Peripheral blood from recipients was analyzed by flow cytometry at 4, 12, and 20 weeks after transplantation. (**J**) Representative FACS profiles of donor-type (CD45.2) and recipient-type (CD45.1) blood cells at 4 weeks post transplantation. (**K**) Dramatic reduction of donor-type (CD45.2) blood cells in recipients transplanted with $Rps27l^{-/-}$ fetal livers. The percentages of CD45.2⁺ cells in peripheral blood at 4, 12, and 20 weeks post transplantation were summarized. $Rps27l^{+/+}$ or $Rps27l^{-/-}$ fetal liver cells were injected into recipient mice (n = 3, for each genotype). Shown are mean ± SD. **p < 0.01, ***p < 0.001, as compared to $Rps27l^{+/+}$ counterparts.

The following figure supplement is available for figure 2:

**Figure supplement 1**. Depletion of hematopoietic stem and progenitor cells (HSPCs) upon *Rps27l* disruption.

bone marrow) are the typical phenotypes of p53 activation. Indeed, by immunohistochemistry analysis, we detected more p53 positively stained cells in the fetal livers and bone marrows derived from $Rps27l^{-/-}$ mice than those from $Rps27l^{+/+}$ control littermates (**Figure 3A**). By immunoblotting, we found that both basal and radiation-induced levels of p53 and p53 target protein Puma are higher in fetal livers derived from $Rps27l^{-/-}$ embryos than in those from $Rps27l^{+/+}$ embryos (**Figure 3B**). Likewise, the levels of p53 and Puma are also higher in $Rps27l^{-/-}$ spleen and brain tissues (**Figure 3C,D**). Furthermore, we observed a moderate increase in the levels of p53 and its two well-known targets, Mdm2 and p21 in MEFs derived from $Rps27l^{-/-}$ embryos than in those from $Rps27l^{+/+}$ embryos (**Figure 3E**). Finally, the p53 levels, induced by (a) ribosomal stress inducer, actinomycin D (Act D), (b) DNA damaging agent, etoposide (**Figure 3—figure supplement 1A**), or (c) ionizing radiation (**Figure 3—figure supplement 1B**), were also higher in $Rps27l^{-/-}$ MEFs than in $Rps27l^{+/+}$ MEFs. It is worth noting that increased p53 in $Rps27l^{-/-}$ MEFs is unlikely due to enhanced DNA damage response, given the similar phosphorylation levels of γH2AX and Chk1 between MEFs of the two genotypes (**Figure 3—figure supplement 1B**). Taken together, our results clearly showed that *Rps27l* disruption causes an increase in p53 levels in multiple organs and cell types, and that increased p53 is transcriptionally active to induce the expression of its downstream targets, namely p21, Mdm2 and Puma.

## Increased p53 level upon *Rps27l* disruption is due to reduced p53 ubiquitylation and degradation

We next determined the mechanism by which *Rps27l* disruption causes p53 increase. Quantitative RT-PCR analysis using four independent sets of MEFs revealed that *Rps27l* disruption did not increase p53 mRNA level, but as expected, increased mRNA levels of two p53 targets, Mdm2 and p21, due to p53 transactivation (**Figure 4—figure supplement 1A**). The [³⁵S]-methionine labeling experiment in two independent pairs of MEFs showed that the rate of p53 synthesis is similar regardless of *Rps27l* status (**Figure 4—figure supplement 1B**), indicating that *Rps27l* disruption does not alter the p53 protein synthesis. We then focused our attention on the rate of p53 degradation by measuring the p53 protein half-life. We found that p53 half-life was doubled from ~10 min to ~25 min in MEFs upon *Rps27l* disruption (**Figure 4A**) due to reduced ubiquitylation and degradation (**Figure 4B**, **Figure 4—figure supplement 1C**). Because the major E3 ligase for p53 degradation is Mdm2, we performed an in vitro ubiquitylation assay, using, as the E3 source, the Mdm2 complex immuno-precipitated from the paired $Rps27l$ MEFs, and found that the p53 ubiquitylation was reduced in $Rps27l^{-/-}$ MEFs substantially (**Figure 4C**), indicating a reduced Mdm2 ligase activity toward p53.

## *Rps27l* disruption alters the levels and complex formation of Mdm2 and Mdm4

We have shown that *Rps27l* disruption increases the levels of both Mdm2 mRNA (**Figure 4—figure supplement 1A**) and protein (**Figure 3C–E**). We then measured the Mdm2 protein half-life and Mdm2 self-ubiquitylation in $Rps27l^{+/+}$ vs $Rps27l^{-/-}$ MEFs and found that *Rps27l* disruption extends Mdm2 protein half-life from ~18 min to 40 min (**Figure 4A**), which is likely attributable to a decreased

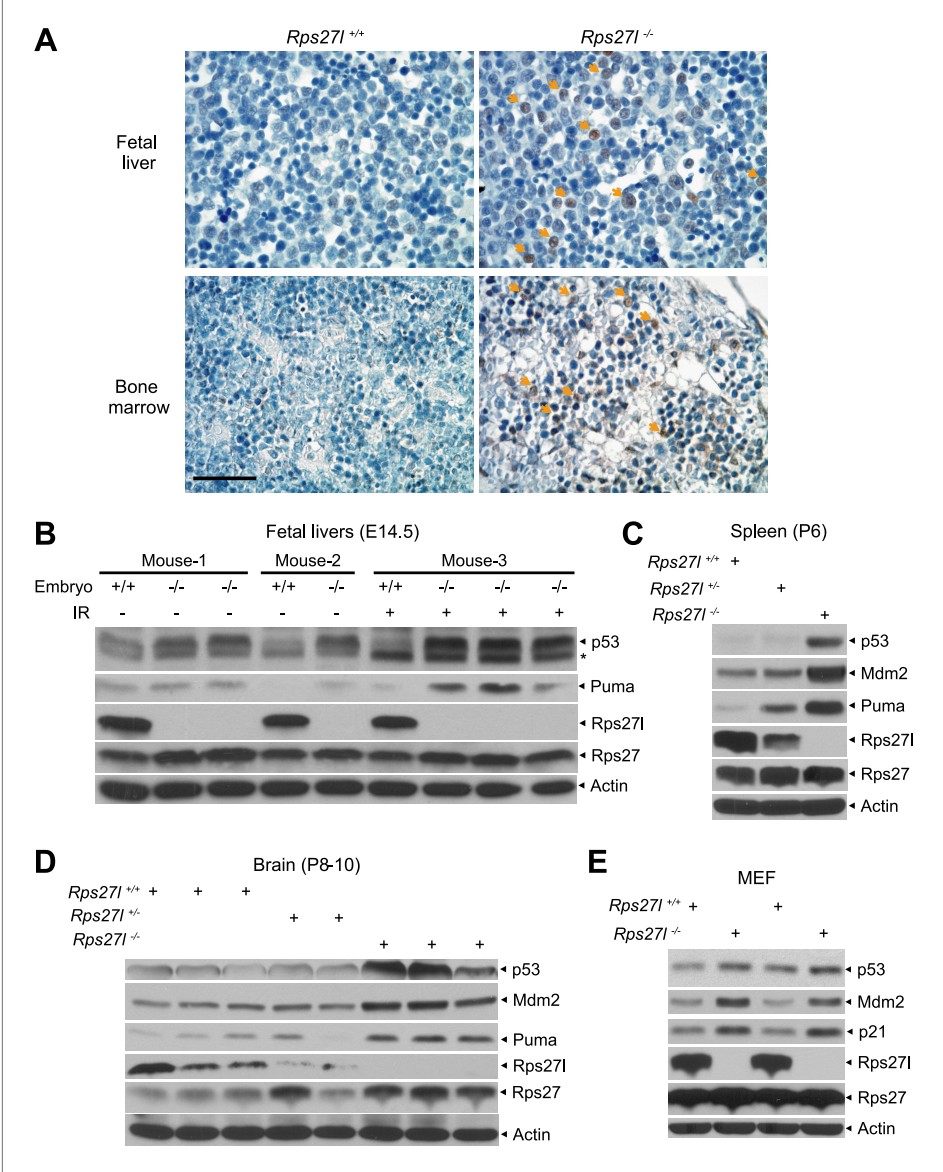

**Figure 3**. *Rps27l* disruption causes a moderate increase of p53 and p53 targets. (**A**) Representative p53 staining in fetal livers and bone marrows. Fetal livers at E14.5 and bone marrows at P6 were immuno-stained with p53 Ab. Arrows point to p53 positive staining. Scale bar represents 40 µm. (**B**) Accumulation of p53 and Puma in *Rps27l*⁻/⁻ fetal livers. Fetal livers isolated from embryos of non-irradiated or irradiated pregnant *Rps27l*⁺/⁻ females (E14.5) were lysed for IB at 5 hr after ionizing radiation at 5 Gy. * denotes a nonspecific band. (**C**) Accumulation of p53 and p53 targets in *Rps27l*⁻/⁻ spleens. Several spleens with the same genotype from P6 pups were harvested, pooled, homogenized, and subjected to IB. (**D**) Accumulation of p53 and p53 targets in *Rps27l*⁻/⁻ brains. Brains from P8–10 pups were harvested and lysed for IB. (**E**) Accumulation of p53 and p53 target proteins in *Rps27l*⁻/⁻ MEFs. Two independent pairs of MEFs from embryos at E13.5 were lysed for IB with indicated antibodies.

The following figure supplement is available for figure 3:

**Figure supplement 1**. Increased p53 levels in responsive to various stresses in *Rps27l*⁻/⁻ MEFs.

Mdm2 self-ubiquitylation (***Figure 4D***). We further used p53-null H1299 cells to determine this RPS27L-MDM2 inverse relationship in a p53-independent manner. Consistently, we found that ectopic expression of RPS27L decreases MDM2 protein in a dose dependent manner, whereas RPS27L knockdown increases MDM2 protein (***Figure 4E***). Furthermore, ectopic RPS27L expression shortens the MDM2

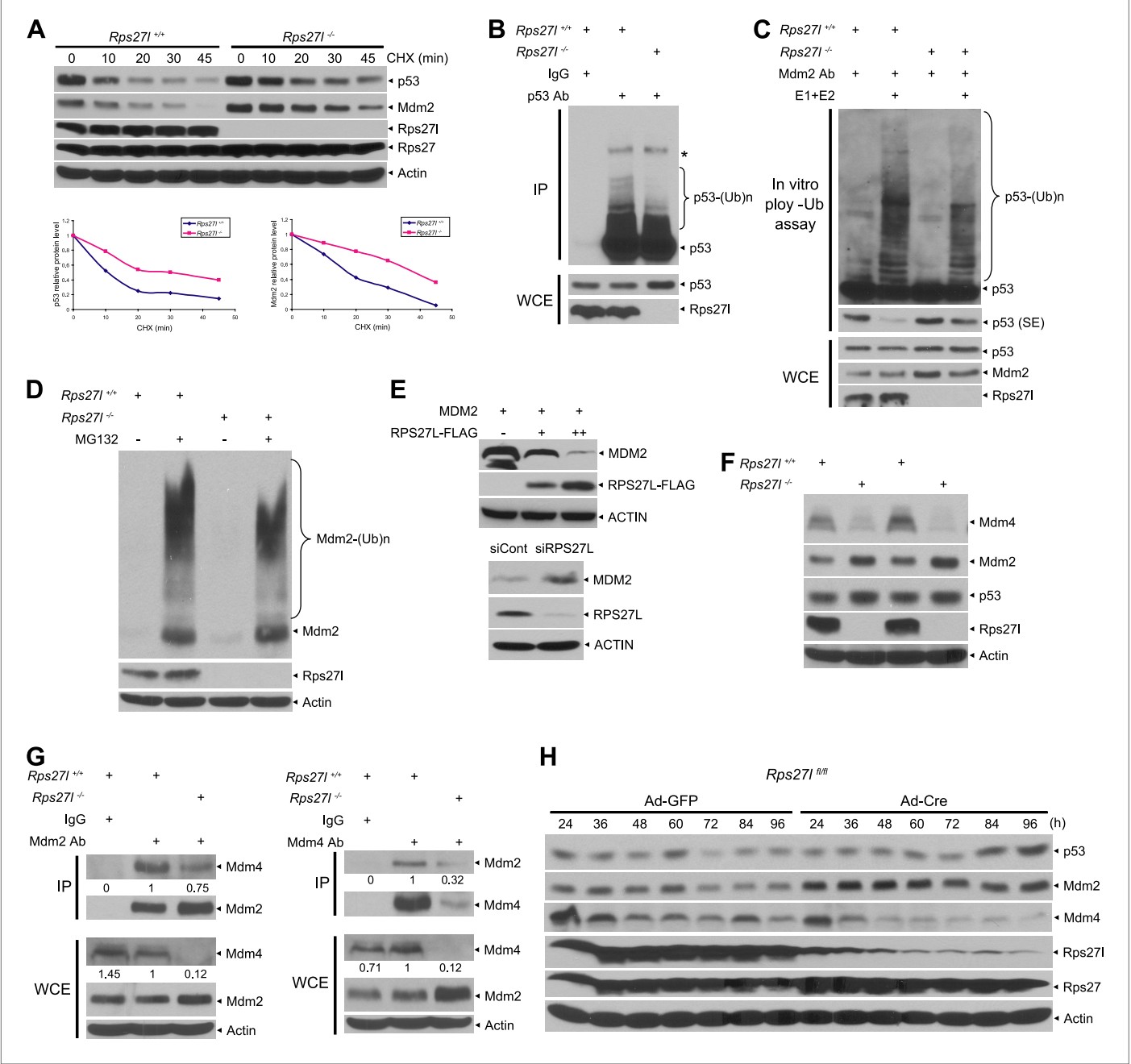

**Figure 4**. *Rps27l* disruption alters the levels of the p53-Mdm2-Mdm4 axis. (**A**) Extension of protein half-lives of p53 and Mdm2 upon *Rps27l* disruption. *Rps27l*[+/+] or *Rps27l*[−/−] MEFs were harvested at various time points post CHX treatment for IB (top). Densitometry quantification was performed with ImageJ, and the decay curves are shown (bottom). (**B**) *Rps27l* disruption impairs the ubiquitylation of endogenous p53 in vivo. *Rps27l*[+/+] or *Rps27l*[−/−] MEFs were harvested after 4 hr of MG132 treatment for IP with p53 Ab or normal IgG control, followed by IB with p53 Ab (top), or for direct IB with p53 or Rps27l Ab (bottom). * denotes a nonspecific band. WCE: whole cell extract. (**C**) *Rps27l* disruption impairs p53 ubiquitylation in vitro. The Mdm2-Mdm4 E3 and p53 (substrate) complex was immunoprecipitated with Mdm2 Ab from MG132 treated *Rps27l*[+/+] or *Rps27l*[−/−] MEFs, and added into an in vitro ubiquitylation reaction mixture containing ATP, ubiquitin, E1, and E2 (UbcH5b). After 60 min incubation with continuous vortexing, the reaction mixture was subjected to IB using p53 Ab. SE: short exposure. (**D**) Reduced Mdm2 self-ubiquitylation upon *Rps27l* depletion. *Rps27l*[+/+] or *Rps27l*[−/−] MEFs were treated with MG132 for 4 hr before being harvested for IB using indicated Abs. (**E**) Negative regulation of MDM2 protein levels by RPS27L. p53-null H1299 lung cancer cells were transfected with MDM2 alone, or in combination with two concentrations of FLAG-tagged RPS27L for 48 hr (top), or infected with lenti-virus targeting RPS27L or scrambled control siRNA for 3 days (bottom), followed by IB with indicated Abs. (**F**) Reduced Mdm4 protein level in *Rps27l*[−/−] MEFs. Two independent pairs of MEFs were harvested and subjected to IB with indicated Abs. (**G**) Reduced Mdm2-Mdm4 complex in *Rps27l*[−/−] MEFs. Whole cell extracts of *Rps27l*[+/+] or *Rps27l*[−/−] MEFs were subjected to IP with Mdm2 Ab or normal IgG control, followed by IB

*Figure 4. Continued on next page*

*Figure 4. Continued*

with Mdm4 Ab (left), or subjected to IP with Mdm4 Ab or normal IgG control, followed by IB with Mdm2 Ab (right). WCE were also subjected to direct IB with indicated Abs. Densitometry quantification was performed with ImageJ. (**H**) Sequential change in the protein levels of Mdm2, Mdm4, and p53 upon acute depletion of *Rps27l*. *Rps27l^fl/fl^* MEFs were harvested at various time points following adenoviral infection and subjected to IB.

The following figure supplement is available for figure 4:

**Figure supplement 1**. Rps27l regulates p53 ubiquitylation and Mdm2 protein half-life.

protein half-life, while RPS27L knockdown extends it (*Figure 4—figure supplement 1D*). Thus, *Rps27l* disruption stabilizes Mdm2 by extending its protein half-life likely via reducing its self-ubiquitylation.

It appears paradoxical that an increased Mdm2 leads to a decreased p53 ubiquitylation and degradation. Given that Mdm2 is known to bind to its family member, Mdm4, to form the most active heterodimer E3 ligase toward p53 (*Tanimura et al., 1999*; *Kostic et al., 2006*; *Kawai et al., 2007*), we determined whether reduced Mdm2 E3 toward p53 upon *Rps27l* disruption is due to a reduced amount of Mdm4. Indeed, total cellular levels of Mdm4 were significantly reduced in several independent isolates of *Rps27l^−/−^* MEFs, as compared to the *Rps27l^+/+^* littermates (*Figure 4F*). We then determined the binding affinity of Mdm2 and Mdm4 by the two-reciprocal immunoprecipitation (IP) assay in *Rps27l^+/+^* vs *Rps27l^−/−^* MEFs under unstressed native condition in the absence of proteasome inhibitor MG132. Although the total cellular level of Mdm4 was much lower in *Rps27l^−/−^* than in *Rps27l^+/+^* MEFs with a ratio of 0.12 vs 1, the level of Mdm2-bound Mdm4 (pulled-down by Mdm2 IP) was much higher with a ratio of 0.75 vs 1 (*Figure 4G*, left). Reciprocally (Mdm4 IP), the level of Mdm4-bound Mdm2 was also higher with a ratio of 0.32 vs 1 (*Figure 4G*, right). Thus, although a much lower total levels of Mdm4 in *Rps27l^−/−^* MEFs, most of Mdm4 molecules were found in the complex with Mdm2. These results clearly demonstrated that in the absence of Rps27l, Mdm2 actually has a higher binding affinity towards Mdm4, which might facilitate Mdm2-mediated Mdm4 degradation, leading to a decreased level of Mdm4, compromised Mdm2-Mdm4 ligase activity towards p53, and consequent p53 accumulation.

To elucidate the complicated interplays among p53-Mdm2-Mdm4-Rps27l, we generated *Rps27l^fl/fl^* mice (unpublished data) and determined the initial and/or sequential event(s) that cause(s) the change in the Mdm2-Mdm4-p53 proteins upon acute depletion of *Rps27l*. *Rps27l^fl/fl^* MEFs were infected with adenovirus expressing Cre recombinase (Ad-Cre) to deplete *Rps27l*. Compared to the Ad-GFP controls, acute depletion of *Rps27l* caused an elevated Mdm2, starting at 24 hr post Ad-Cre infection and lasting up to 96 hr, followed by depletion of Mdm4, starting at 48 hr. A moderate accumulation of p53 was not seen until the later time points, starting at 72–84 hr following Ad-Cre infection (*Figure 4H*). Furthermore, Mdm2 accumulation, followed by Mdm4 reduction, is independent of p53, since a similar changing pattern was seen when *Rps27l^fl/fl^;Trp53^−/−^* MEFs were infected with Ad-Cre (*Figure 4—figure supplement 1E*). Taken together, these results clearly demonstrated that upon *Rps27l* depletion (not necessary for complete elimination), Mdm2 increases first, followed by an Mdm4 decrease, and finally a p53 increase. These sequential changes supported the notion that *Rps27l* depletion somehow stabilizes Mdm2, which promotes Mdm4 degradation, as also shown previously (*de Graaf et al., 2003*; *Kawai et al., 2003*; *Pan and Chen, 2003*), leading to a suboptimal Mdm2-Mdm4 complex with reduced ligase activity toward p53, and eventually p53 accumulation.

## Rps27l is a ribosomal protein whose depletion triggers ribosomal stress to stabilize Mdm2

By amino sequence comparison, Rps27l is a family member of ribosomal protein S27 (Rps27) (*Xiong et al., 2011*). However, it has never been determined previously whether Rps27l is indeed a ribosomal protein. By ribosomal profiling we found that like its family member Rsp27, Rps27l is localized in the ribosomes, along with other known Mdm2-binding ribosomal proteins, including Rpl5, Rpl11, and Rps7, and that *Rps27l* disruption does not cause obvious alterations in ribosomal profile, given the profiling patterns are largely overlapping with each other, except for the 40S peak, which is relatively lower in *Rps27l^−/−^* MEFs (*Figure 5A*).

We next determined whether *Rps27l* disruption causes aberrant ribosomal assembly. We first used qRT-PCR analysis and found that *Rps27l* disruption had no effect on the levels of 45S rRNA (*Figure 5—figure supplement 1A*), indicating Rps27l loss does not impair the transcription of rDNA. Ethidium

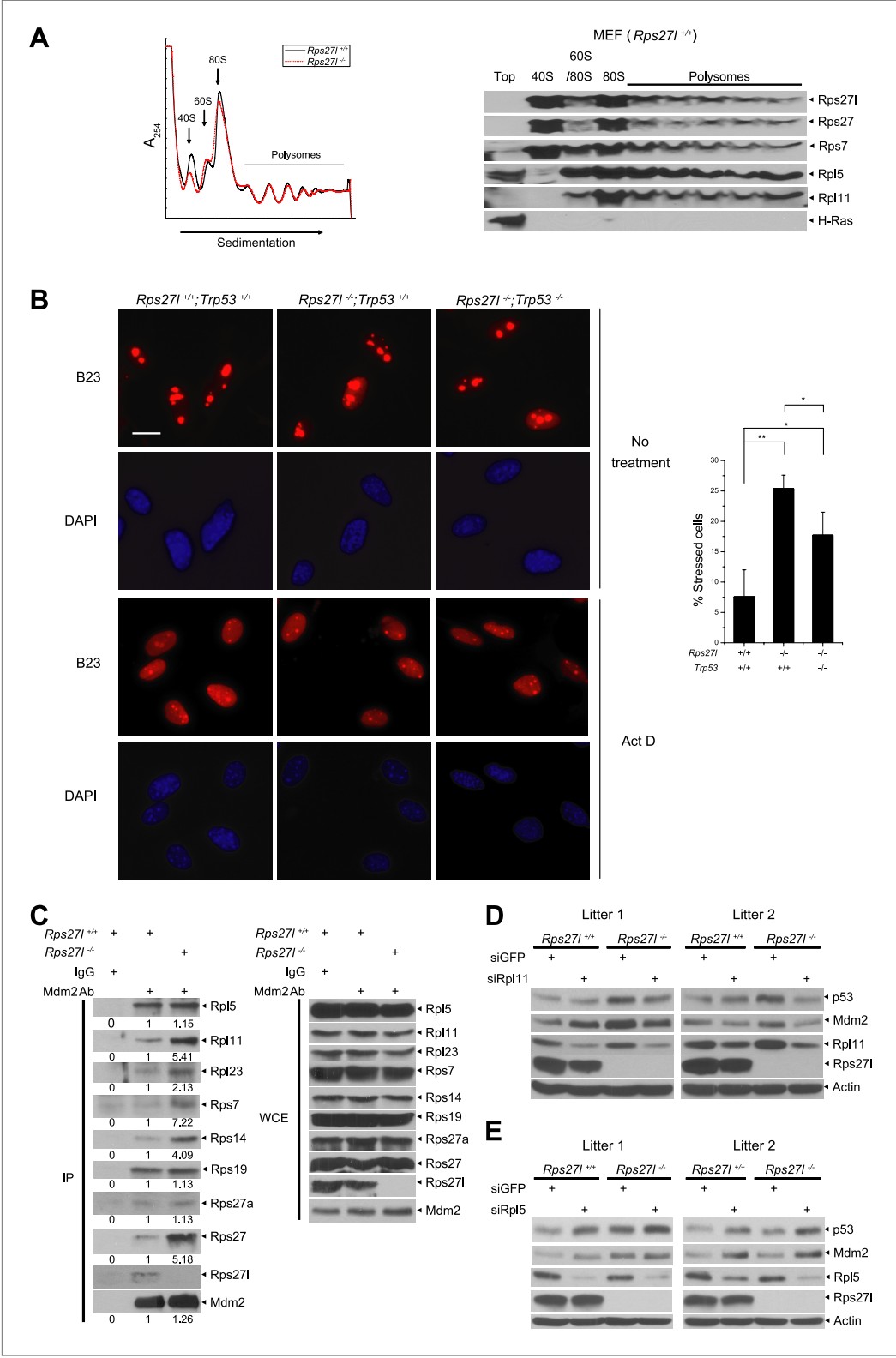

**Figure 5**. Rps27l is a ribosomal protein whose depletion triggers ribosomal stress. (**A**) Rps27l is a ribosomal protein. *Rps27l⁺/⁺ or Rps27l⁻/⁻* MEFs were treated with CHX for 30 min before harvesting for ribosomal profiling. The cytoplasmic extracts were loaded on sucrose gradients (10%–50%) and subjected to ultra-centrifugation. Gradients were then fractionated and measured by optical density at 254 nm (top). The fractions from *Rps27l⁺/⁺* MEFs were
*Figure 5. Continued on next page*

*Figure 5. Continued*

subjected to IB using indicated Abs (bottom). (**B**) B23 is released from nucleoli upon Rps27l depletion. MEFs with indicated genotypes were left untreated or treated with 5 nM Act D for 24 hr, followed by immunofluoresent staining of B23 (left). Scale bar represents 20 μm. Cells with nucleolus and/or nucleoplasmic B23 staining were counted and expressed as percentage of total cells (at least 200) counted (right). Shown are mean ± SD. **p < 0.01; *p < 0.05. (**C**) The change in Mdm2 binding of various ribosomal proteins in *Rps27l*⁻/⁻ MEFs. *Rps27l*⁺/⁺ or *Rps27l*⁻/⁻ MEFs were treated with 5 nM Act D for 4 hr before being harvested for IP with Mdm2 Ab or normal IgG control, followed by IB with indicated Abs (left), or for direct IB with indicated Abs (right). Densitometry quantification was performed with ImageJ. (**D** and **E**) Activation of p53 by *Rps27l*-deficiency is dependent on Rpl11, but not Rpl5. Two independent pairs of MEFs were infected with lentivirus expressing short hairpin RNA (shRNA) against GFP as a negative control or against Rpl11 (**D**) or Rpl5 (**E**) before being harvested for IB with indicated Abs.

The following figure supplement is available for figure 5:

**Figure supplement 1**. *Rps27l* disruption has minimal effects on rDNA transcription, rRNA processing and/or protein synthesis.

bromide staining also showed a similar level of mature forms of 28S and 18S rRNAs (***Figure 5—figure supplement 1B***). We then performed pulse-chase labeling of rRNA and found that *Rps27l* loss caused a moderate decrease in the kinetics of rRNA processing from 47S to 18S (***Figure 5—figure supplement 1C,D***). Finally, we found that Rps27l loss had no effect on global protein synthesis, as determined by a [³⁵S]-methionine pulse-chase experiment (***Figure 5—figure supplement 1E,F***). Taken together, we conclude that p53 induction upon *Rps27l* disruption is unlikely due to aberrant protein synthesis, but reduced rRNA processing may contribute to some extent.

It is established that upon ribosomal stress a number of ribosomal proteins are released from nucleoli to bind to Mdm2, leading to its stabilization (***Zhang and Lu, 2009***). We then determined if *Rps27l* disruption triggers ribosomal stress by staining MEFs with B23/NPM, a nucleolus protein known to be released to nucleoplasm upon ribosomal stress, as a marker (***Jin et al., 2004***). While only ~7% of *Rps27l*⁺/⁺ MEFs had nucleoplasm staining, ~25% of *Rps27l*⁻/⁻ MEFs had positive nucleoplasm staining (p <0.01) (***Figure 5B***). Positive control Act D caused nucleolar B23 release in nearly all of MEFs, regardless of *Rps27l* status (***Figure 5B***). Since p53 has been shown to constrain the ribosome biogenesis (***Golomb et al., 2014***), we determined potential involvement of p53 in the process and found that nucleolar B23 release was also significantly increased in *Rps27l*⁻/⁻;*Trp53*⁻/⁻ MEFs, although to a lesser extent (7 vs 17%) (***Figure 5B***). Thus, depletion of *Rps27l* triggers ribosomal stress to cause B23 release from nucleoli, which is largely independent of p53.

We further determined the levels of Mdm2-bound ribosomal proteins as an independent marker for ribosomal stress (***Zhang and Lu, 2009***; ***Zhou et al., 2012***). Although there is no measurable difference between *Rps27l*⁺/⁺ and *Rps27l*⁻/⁻ MEFs (two independent pairs) in total cellular levels of several ribosomal proteins known to bind to Mdm2, including Rpl5, Rpl11, Rpl23, Rps7, Rps14, Rps19 (***Figure 5—figure supplement 1G***), we did detect in *Rps27l*⁻/⁻ MEFs an increased Mdm2 binding of Rpl11, Rpl23, Rps7, Rps14, and Rps27, but not Rpl5, Rps27a, and Rps19 (***Figure 5C***). Moreover, since recent studies have documented the central role of RPL5 and RPL11 in ribosomal stress-induced p53 activation (***Bursac et al., 2012***; ***Golomb et al., 2014***), we determined whether activation of p53 by *Rps27l*-deficiency is dependent on Rpl5 and Rpl11 by siRNA-based silencing approach. In two pairs of MEFs derived from two independent littermates, silencing of Rpl11 (with enhanced Mdm2 binding upon *Rps27l* disruption), but not Rpl5 (without changing Mdm2 binding) abrogated the p53 activation in *Rps27l*⁻/⁻ MEFs, suggesting activation of p53 by *Rps27l*-deficiency is dependent on Rpl11, but not Rpl5 (***Figure 5D,E***). Interestingly, we reproducibly observed that Rpl5 silencing increased p53 levels in MEFs using three independent Rpl5-targeting shRNAs (***Figure 5E*** and data not shown). Taken together, our results suggest that *Rps27l* disruption triggers ribosomal stress to increase the binding of Mdm2 to selective sets of ribosomal proteins, particularly Rpl11, leading to Mdm2 stabilization.

### *Rps27l* disruption destabilizes Mdm4 via Mdm2-mediated ubiquitylation and degradation

We then addressed mechanistically how stabilized Mdm2 failed to promote p53 degradation in *Rps27l*⁻/⁻ MEFs with focus on Mdm4. We found that substantial reduction of Mdm4 in *Rps27l*⁻/⁻ MEFs

was not due to decreased Mdm4 mRNA transcription (*Figure 6—figure supplement 1A*), nor decreased Mdm4 mRNA translation (data not shown), but enhanced degradation, since treatment with proteasome inhibitor, MG132 for 4 hr caused its accumulation (*Figure 6—figure supplement 1B*). It is noteworthy that MG132-induced accumulation of Mdm2 and p53 is much more substantial, given that both Mdm2 and p53 have a much shorter protein half-life.

Consistent with the fact that Mdm2 promotes Mdm4 ubiquitylation and degradation (*de Graaf et al., 2003*; *Kawai et al., 2003*; *Pan and Chen, 2003*) under some stressed conditions including ribosomal stress (*Gilkes et al., 2006*), we found Mdm4 polyubiquitylation was indeed increased with accompanied reduction of Mdm4 levels in *Rps27l*[−/−] MEFs (*Figure 6A*). Furthermore, in p53-null H1299 cells, MDM2-mediated MDM4 polyubiquitylation was substantially inhibited by ectopically expressed RPS27L (*Figure 6B*). A similar result was seen in 293 cells (*Figure 6—figure supplement 1C*). Consistently, the Mdm4 protein half-life was significantly shortened upon *Rps27l* disruption in MEFs (*Figure 6C*) or upon RPS27L knockdown in H1299 cells (*Figure 6—figure supplement 1D*), but significantly extended upon ectopic expression of RPS27L in H1299 cells (*Figure 6—figure supplement 1E*). Taken together,

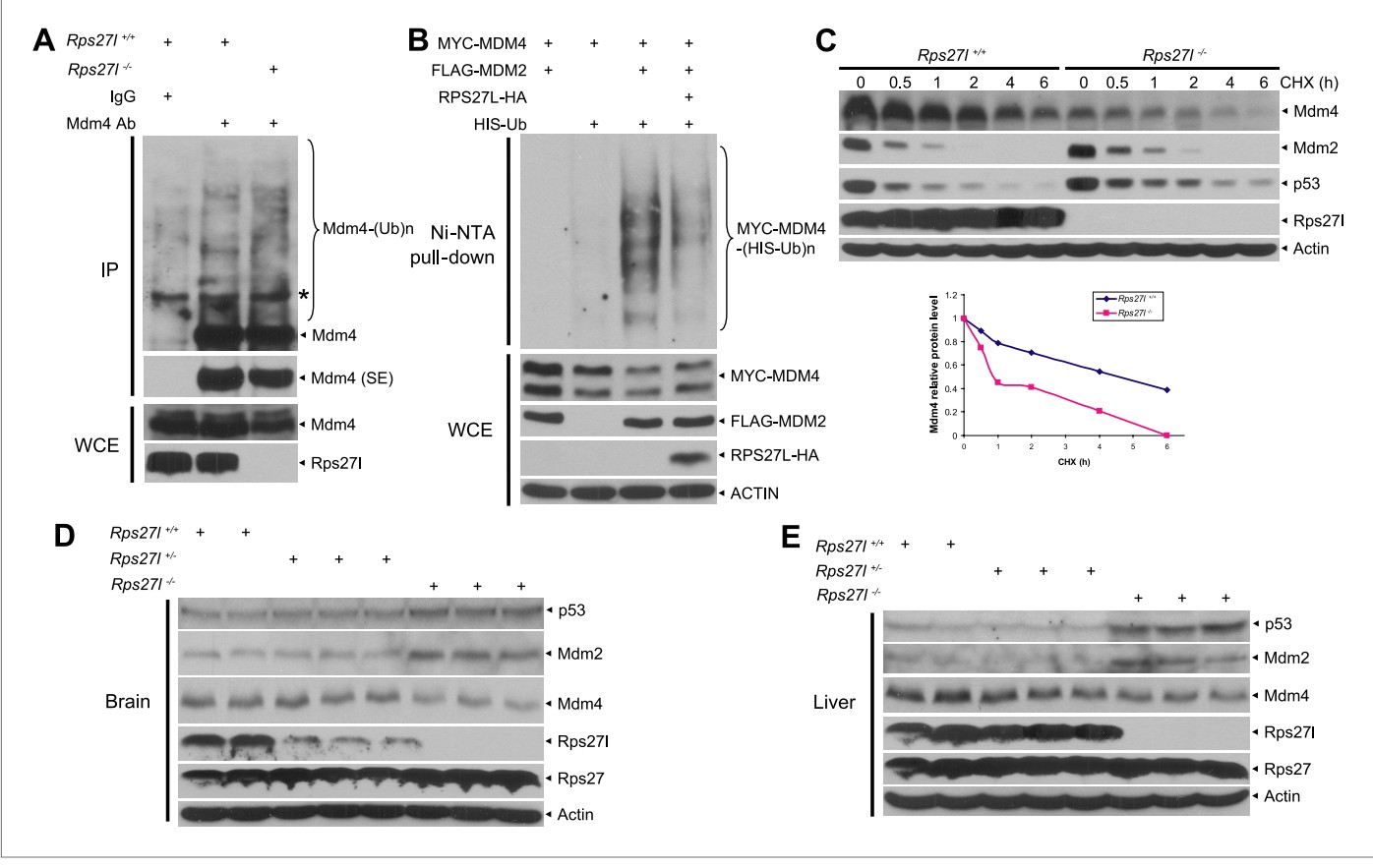

**Figure 6**. *Rps27l* depletion leads to Mdm2-mediated Mdm4 degradation. (**A**) Increased Mdm4 ubiquitylation upon *Rps27l* disruption. *Rps27l*[+/+] or *Rps27l*[−/−] MEFs were treated with MG132 before being harvested for IP with Mdm4 Ab or normal IgG control, followed by IB with Mdm4 Ab (top), or for direct IB with indicated Abs (bottom). SE: short exposure. (**B**) MDM2-mediated MDM4 ubiquitylation is inhibited by ectopic RPS27L. H1299 cells were transfected with indicated plasmids. At 24 hr post-transfection, cells were treated with MG132 for 4 hr, and then harvested for purification of His-tagged ubiquitylated proteins by Ni-NTA beads, followed by IB with Myc-tag Ab to detect MDM4 ubiquitylation (top), or for direct IB with indicated Abs (bottom). (**C**) Shortened Mdm4 protein half-life upon *Rps27l* disruption. *Rps27l*[+/+] or *Rps27l*[−/−] MEFs were harvested at various time points post CHX treatment for IB (top). Densitometry quantification was performed with ImageJ, and the decay curves are shown (bottom). (**D** and **E**) Increased protein levels of p53 and Mdm2 and decreased protein level of Mdm4 in *Rps27l*[−/−] brains (**D**) and livers (**E**). Brain and liver tissues from P6 pups with indicated genotypes were harvested and lysed for IB.

The following figure supplement is available for figure 6:

**Figure supplement 1**. Rps27l regulates Mdm2-mediated Mdm4 degradation.

our study demonstrated that stabilized Mdm2 upon *Rps27l* depletion indeed promotes ubiquitylation and degradation of Mdm4 to reduce its cellular levels. Finally, we found a consistent increase of p53 and Mdm2 and a decrease of Mdm4 in several independent *Rps27l*-null tissues of brain and liver (**Figure 6D,E**).

## Simultaneous deletion of *Trp53* rescues postnatal death and hematopoietic failure

We next determined whether increased p53 plays a causal role in developmental defects seen in *Rps27l*⁻/⁻ mice by simultaneous deletion of *Trp53*. Intercrossing of *Rps27l*⁺/⁻;*Trp53*⁺/⁻ mice gave rise to mice with the *Rps27l*⁻/⁻ background in combination with three p53 genotypes. All mice with *Rps27l*⁻/⁻;*Trp53*⁺/⁺ background die within 3 weeks of age with one mouse living up to 35 days (**Figure 1—figure supplement 1E**, **Figure 7A**). Consistently, HSPCs in bone marrow were significantly reduced (**Figure 7—figure supplement 1A,B**), as well as the peripheral blood cells (**Figure 7—figure supplement 1C,D**). In contrast, both *Rps27l*⁻/⁻;*Trp53*⁺/⁻ and *Rps27l*⁻/⁻;*Trp53*⁻/⁻ mice developed normally and are fertile with expected Mendelian distribution (**Figure 7A**), indicating a complete rescue of the postnatal death by either heterozygous or homozygous deletion of *Trp53*. Note that the life-span greater than 5 weeks was used in **Figure 7A**, given that the longest life-span of a *Rps27l*⁻/⁻;*Trp53*⁺/⁺ mouse is 5 weeks (**Figure 1—figure supplement 1E**). At the organ and cellular levels, *Trp53* deletion of either one or both alleles rescued overall defective phenotypes seen in *Rps27l*⁻/⁻ mice as follows: (1) growth retardation, as reflected by the recovery of reduced body size and weight (**Figure 7B,C**); (2) organ specific hypo-cellularity, as reflected by the recovery of total cell numbers in spleen, thymus and bone marrow (**Figure 7D,E**). On the other hand, however, the recovery of bone marrow HSPCs depletion was completely rescued by homozygous but not by heterozygous *Trp53* deletion, since some of HSPCs were only partially recovered in *Rps27l*⁻/⁻;*Trp53*⁺/⁻ mice (**Figure 7F,G**), although it does not affect the survival of mice. Partial to complete recovery was seen in peripheral blood cells (**Figure 7H**) as well as in differentiated lineages from peripheral blood (**Figure 7—figure supplement 1E**).

We further determined whether *Trp53* deletion rescues hematopoietic failure of *Rps27l*⁻/⁻ fetal liver. Indeed, the percentage of HSPCs was significantly increased upon deletion of one or both alleles of *Trp53* (**Figure 7I,J**). More importantly, in a non-competitive reconstitution assay, inability of *Rps27l*⁻/⁻ fetal liver cells to reconstitute sterilized bone marrow was rescued by heterozygous or homozygous deletion of *Trp53* (**Figure 7K**). The percentage of donor derived cells in rescued chimeric mice approached the wild type control (**Figure 7—figure supplement 1F**). Thus, p53 increase upon *Rps27l* disruption is fully responsible for phenotypic defects observed in *Rps27l*⁻/⁻ mice.

## Rps27l is required for genomic stability

The long-term animal survival studies revealed that compared to *Rps27l*⁺/⁺;*Trp53*⁻/⁻ mice, *Rps27l*⁻/⁻;*Trp53*⁻/⁻ mice had a similar life-span with ~50% of mice dying mainly of lymphoma at age of ~150 days (**Figure 8A**, p = 0.713, log-rank test), indicating *Rps27l* genotype has no effect on the life-span and lymphomas development in the *Trp53*⁻/⁻ mice. However, compared to *Rps27l*⁺/⁺;*Trp53*⁺/⁻ and *Rps27l*⁺/⁻;*Trp53*⁺/⁻ mice, *Rps27l*⁻/⁻;*Trp53*⁺/⁻ mice had a significantly shortened life-span (**Figure 8B**, p = 0.0036, log-rank test). Whole body necropsy revealed that out of 14 *Rps27l*⁻/⁻;*Trp53*⁺/⁻ mice died within a period of 400 days (50% death rate), seven developed T-lymphoblastic lymphoma (**Figure 8C,D**), one developed T-cell lymphoma detected in tissues of thymus, lymph node and spleen (**Figure 8—figure supplement 1A**), and one developed B-cell lymphoma seen in a much enlarged lymph node (**Figure 8—figure supplement 1B**). The remaining mice had enlarged liver and/or spleen (not shown). The results indicate that *Rps27l* disruption accelerated the formation of spontaneous lymphomas under the *Trp53*⁺/⁻ background. We then collected more lymphoma tissues from *Rps27l*⁻/⁻;*Trp53*⁺/⁻ mice at age of 4–6 months for *Trp53* genotyping and found that the wild type *Trp53* allele was deleted in 29 out of 30 (97%) lymphomas genotyped (**Figure 8—figure supplement 1C,D**), indicating that *Rps27l* disruption imposed the selection pressure against wild-type *Trp53*. We further measured the genome integrity of these lymphoma cells by metaphase chromosome spread and found a high degree of aneuploidy in up to 63% of total population (**Figure 8E**, **Figure 8—figure supplement 1E**). Centromere–centromere fusions were also seen in some diploid lymphoma cells (**Figure 8E**, arrow). Thus, *Rps27l* disruption causes genomic instability and *Trp53* deletion, eventually leading to lymphoma development. To determine which event (genomic instability vs *Trp53* deletion) occurs first, we performed

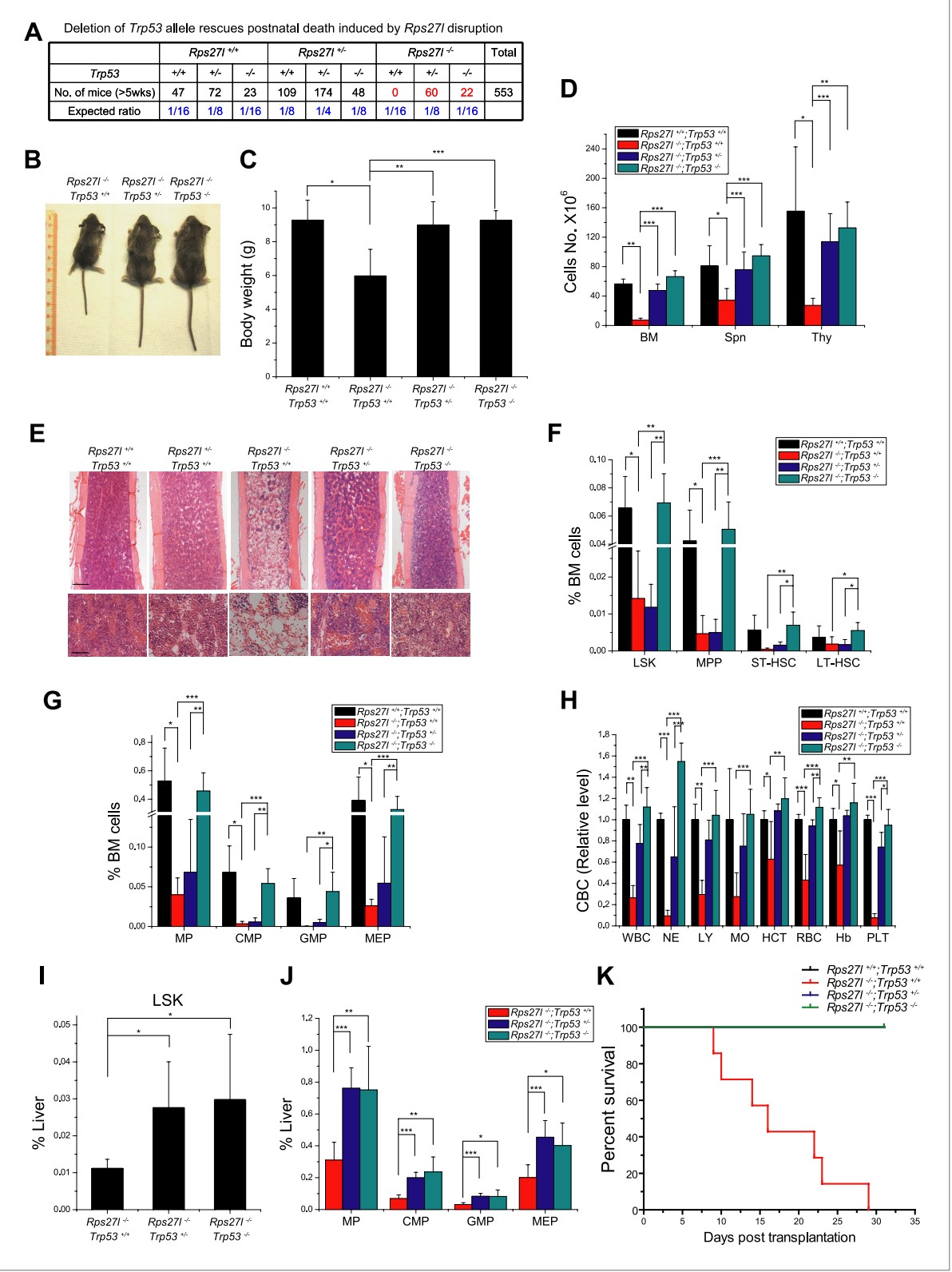

**Figure 7**. Simultaneous deletion of *Trp53* rescues growth retardation and HSPCs depletion. (**A**) Deletion of *Trp53* rescues postnatal death by *Rps27l* disruption. Lower than expected number of mice with *Trp53⁻/⁻* genotype (regardless of *Rps27l* genotype) is due to high frequency of developmental abnormalities during embryonic and neonatal stages which cause the premature death (***Armstrong et al., 1995***; ***Sah et al., 1995***). (**B**–**D**) Deletion of *Trp53* rescues growth retardation and organ hypocellularity. Representative mice at P18 of three genotypes were photographed (**B**). The bodies (**C**) were

*Figure 7. Continued on next page*

Figure 7. Continued

weighed; and the total cell numbers (**D**) of bone marrow (femur and tibia from two hind limbs), spleen, and thymus were counted from P18 mice with genotypes of $Rps27l^{+/+};Trp53^{+/+}$ (n = 3), $Rps27l^{-/-};Trp53^{+/+}$ (n = 7), $Rps27l^{-/-};Trp53^{+/-}$ (n = 10), $Rps27l^{-/-};Trp53^{-/-}$ (n = 5). Shown are mean ± SD. *p < 0.05, **p < 0.01, and ***p < 0.001. (**E**) Representative H&E staining of bone marrows in femurs from P18 mice. Scale bars represent 200 μm (top) or 40 μm (bottom). (**F** and **G**) Deletion of $Trp53$ rescues HSPCs depletion in $Rps27l^{-/-}$ bone marrow. The percentage of LSK, MPP, ST-HSC, and LT-HSC (**F**); and the percentage of MP, CMP, GMP, and MEP (**G**) in bone marrow from P18 mice with genotypes of $Rps27l^{+/+};Trp53^{+/+}$ (n = 4), $Rps27l^{-/-};Trp53^{+/+}$ (n = 5), $Rps27l^{-/-};Trp53^{+/-}$ (n = 7), and $Rps27l^{-/-};Trp53^{-/-}$ (n = 5). LSK: Lin$^-$/Sca-1$^-$/c-Kit$^+$; MPP: Lin$^-$/Sca-1$^-$/c-Kit$^+$/CD48$^+$/CD150$^-$; ST-HSC: Lin$^-$/Sca-1$^-$/c-Kit$^+$/CD48$^+$/CD150$^+$; LT-HSC: Lin$^-$/Sca-1$^-$/c-Kit$^+$/CD48$^-$/CD150$^+$. Shown are mean ± SD. *p < 0.05, **p < 0.01, and ***p < 0.001. (**H**) Deletion of $Trp53$ rescues defects in $Rps27l^{-/-}$ peripheral blood. CBC classification of peripheral blood from $Rps27l^{+/+};Trp53^{+/+}$ (n = 3), $Rps27l^{-/-};Trp53^{+/+}$ (n = 7), $Rps27l^{-/-};Trp53^{+/-}$ (n = 10), $Rps27l^{-/-};Trp53^{-/-}$ (n = 5) mice at P18 was performed. WBC, white blood cells; NE, neutrophils; LY, lymphocytes; MO, monocytes; HCT, hematocrit; RBC, red blood cells; Hb, hemoglobin; PLT, platelets. Shown are mean ± SD. *p < 0.05, **p < 0.01, and ***p < 0.001. (**I** and **J**) Deletion of $Trp53$ rescues HSPCs depletion in $Rps27l^{-/-}$ fetal livers. Flow cytometry analysis was performed to measure the percentage of HSPCs including LSK (**I**), MP, CMP, GMP, and MEP (**J**) in E14.5 fetal livers with genotypes of $Rps27l^{-/-};Trp53^{+/+}$ (n = 5), $Rps27l^{-/-};Trp53^{+/-}$ (n = 7), and $Rps27l^{-/-};Trp53^{-/-}$ (n = 6). Shown are mean ± SD. *p < 0.05, **p < 0.01, and ***p < 0.001, as compared to $Rps27l^{-/-};Trp53^{+/+}$. (**K**) Kaplan–Meier survival curves of recipient mice after transplantation. Fetal liver cells (2 × 10$^6$ cells) from E14.5 embryos with indicated genotypes were respectively injected into lethally irradiated recipient mice (n = 7 for each genotype). p < 0.0001.
The following figure supplement is available for figure 7:

**Figure supplement 1**. Simultaneous deletion of $Trp53$ rescues defective phenotypes caused by $Rps27l$ disruption.

the same metaphase chromosome spread in four independent pairs of early passage primary MEFs and found a significant higher rate of chromosomal aneuploidy in $Rps27l^{-/-};Trp53^{+/-}$ cells than in $Rps27l^{+/+};Trp53^{+/-}$ cells (30% vs 15%) (**Figure 8F**), although wild-type $Trp53$ was retained, as measured by qPCR (**Figure 8G**). Thus, $Rps27l$ disruption triggers genomic instability prior to $Trp53$ deletion and the cells with subsequent $Trp53$ deletion are selected, which outgrow to form spontaneous lymphoma. We further performed the same metaphase chromosome spread in MEFs with the genotype of $Rps27l^{-/-};Trp53^{+/+}$ (n = 7) vs $Rps27l^{+/+};Trp53^{+/+}$ (n = 3) and found a same low rate of aneuploidy at ~10% in both group of MEFs (**Figure 8H**). Thus, $Rps27l$ disruption triggers genomic instability only when one allele of $Trp53$ is deleted or mutated.

Our results implied that $Rps27l$ depletion is responsible for genomic instability by a mechanism that appears to be independent of p53 loss. To further test this, we measured the levels of p53 and its upstream and downstream regulators in three independent pairs of MEFs of $Rps27l^{-/-};Trp53^{+/-}$ vs the littermates of $Rps27l^{+/+};Trp53^{+/-}$, and found that in every case, both basal and induced (by ActD) levels of p53 and p53 downstream (p21 and Mdm2) are higher in $Rps27l$-null MEFs than in $Rps27l$-wt MEFs, whereas the p19/Arf level was lower in the former (**Figure 8I,J**). These results demonstrated that $Rps27l$ disruption induces genomic instability under $Trp53^{+/-}$ background, which triggers p53 to balance the genome, and is indeed independent of p53 loss. On the other hand, $Rps27l$ disruption also confers a selective pressure against p19/Arf, a p53 upstream regulator, although the underlying mechanism is unclear. Nevertheless, p19/Arf reduction would likely contribute to observed Mdm2 increase to degrade Mdm4, leading to p53 accumulation. Finally, it is worth noting that p53 level and activity were higher in $Rps27l^{-/-};Trp53^{+/+}$ than in $Rps27l^{-/-};Trp53^{+/-}$ MEFs (**Figure 8—figure supplement 1F**), and this difference, likely extendable to other organs, such as bone marrow, determines the life and death of mice at the postnatal stages.

## Discussion

### *Rps27l* is an essential gene for postnatal development by modulating p53

In this study, we demonstrated, using a gene-trap mouse model that Rps27l, a p53 downstream ribosomal protein and an Mdm2 E3 ligase substrate, is essential for mouse development. $Rps27l$ disruption causes a moderate increase of p53 which is sufficient to deplete HSPCs in fetal livers and bone marrows, eventually to cause the postnatal death. By both noncompetitive and competitive reconstruction assays, we clearly demonstrated that the HSPCs from $Rps27l^{-/-}$ fetal livers were unable to reconstitute sterilized bone marrow. Rescuing of these defects by simultaneous deletion of a single or both alleles of $Trp53$ indicates that p53 plays a causal role and that a moderate p53 increase is responsible for and sufficient to cause the depletion of these HSPCs via inducing apoptosis. It is noting worthy that although, unlike homozygous $Trp53$ deletion, heterozygous $Trp53$ deletion

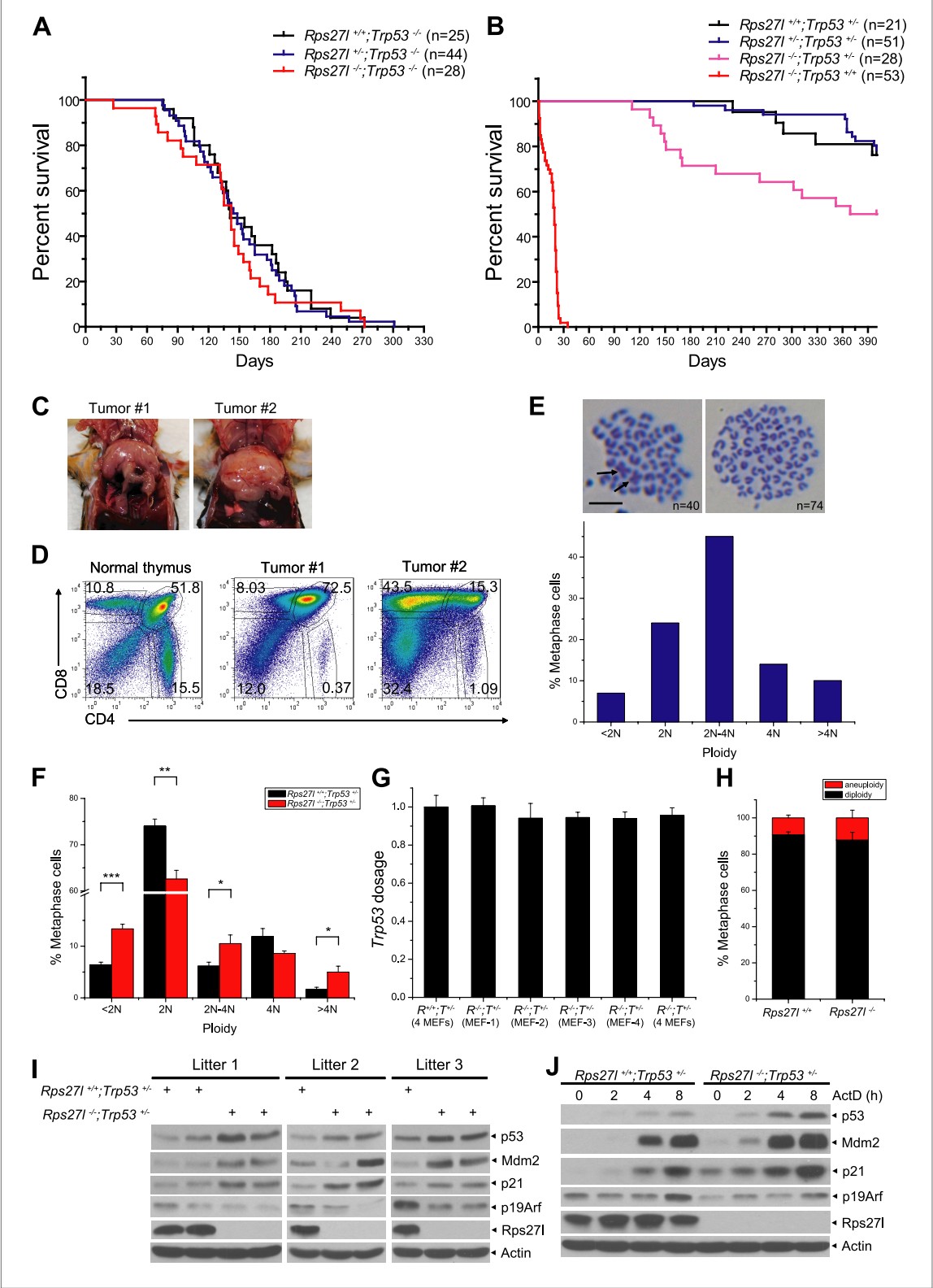

**Figure 8**. *Rps27l* disruption induces genomic instability and spontaneous lymphoma. (**A** and **B**) Kaplan–Meier survival curves of *Trp53*[−/−] (**A**) and *Trp53*[+/−] (**B**) mice with three *Rps27l* genotypes and indicated numbers of mice. p = 0.713 (**A**); p = 0.0036 (**B**). (**C**) Representative pictures of lymphomas developed from *Rps27l*[−/−]; *Trp53*[+/−] mice. (**D**) Representative FACS profiles of T cells from lymphomas. Lymphoma cells were isolated and subjected to *Figure 8. Continued on next page*

*Figure 8. Continued*

FACS analysis using Abs against indicated surface markers. (**E**) Representative pictures of metaphases from four tumors developed from *Rps27l⁻/⁻*; *Trp53⁺/⁻* mice (top). Chromosome numbers (n) were counted. Scale bar represent 10 µm. Frequency of diploid and aneuploidy from four tumors developed in *Rps27l⁻/⁻;Trp53⁺/⁻* mice with 100 metaphases counted (bottom). (**F**) Frequency of diploid and aneuploidy from four pairs of primary MEFs at P2 and P3 derived from *Rps27l⁻/⁻;Trp53⁺/⁻* vs *Rps27l⁺/⁺;Trp53⁺/⁻* embryos with at least 100 metaphases counted in each sample. Shown are mean ± SD. *p < 0.05, **p < 0.01, and ***p < 0.001. (**G**) *Trp53* dosage in four pairs of primary MEFs at P3 derived from *Rps27l⁻/⁻;Trp53⁺/⁻* (*R⁻/⁻;T⁺/⁻*) vs *Rps27l⁺/⁺;Trp53⁺/⁻* (*R⁺/⁺;T⁺/⁻*) embryos. Amounts of p53 DNA in individual MEFs were quantified by qPCR using three sets of primers for exons 5, 6, and 7. The combined results (n = 3 for each primer set, mean ± SD) were presented with the values from *Rps27l⁺/⁺;Trp53⁺/⁻* MEFs averaged and set as 1. (**H**) Frequency of diploid and aneuploidy from primary MEFs at P2 and P3 derived from *Rps27l⁻/⁻;Trp53⁺/⁺* (n = 7) vs *Rps27l⁺/⁺;Trp53⁺/⁺* (n = 3) individual embryos with at least 100 metaphases counted in each sample. Shown are mean ± SD. p = 0.15. (**I** and **J**) Increased levels of p53 and p53 targets and decreased levels of Arf upon *Rps27l* disruption in *Trp53⁺/⁻* MEFs. Primary MEFs at P2 and P3 derived from *Rps27l⁺/⁺;Trp53⁺/⁻* and *Rps27l⁻/⁻;Trp53⁺/⁻* embryos were left untreated (**I**) or treated with 5 nM ActD for indicated time periods (**J**) before being harvested for IB with indicated Abs.

The following figure supplement is available for figure 8:

**Figure supplement 1**. *Rps27l* disruption induces genomic instability and spontaneous lymphoma.

fails to completely rescue the defects in number or percentage of HSPCs in the bone marrows in p18 weaning mice, it is still sufficient to rescue the postnatal growth retardation and death.

## Rps27l is a ribosomal protein that triggers ribosomal stress upon depletion to induce p53

An increasing list of ribosomal proteins (RPs) has been shown in numerous in vitro cell culture studies to bind to Mdm2 and stabilize p53, thus acting as a p53 activator (*Zhang and Lu, 2009*; *Zhou et al., 2012*). It was also reported that siRNA silencing of some ribosomal proteins, such as L23, triggered ribosomal stress, leading to p53 stabilization and activation (*Dai et al., 2004*; *Jin et al., 2004*). We previously showed that RPS27L is an MDM2 binding protein whose overexpression inhibits MDM2-mediated p53 ubiquitylation, whereas its silencing reduces p53 in some cancer lines (e.g., A549 and SJSA), but not in others (e.g., SY5Y, MCF7, U2OS, HCT116, and RKO) (*Xiong et al., 2011*) (data not shown), suggesting RPS27L might act as a p53 activator in a cancer cell dependent manner. Here we reported that Rps27l is indeed a ribosomal protein, whose depletion triggers ribosomal stress (as evidenced by B23 nucleoplasm staining and enhanced RPs-Mdm2 binding) with moderate reduction in ribosomal RNA processing, but no effect on general protein synthesis. We further found, using *Rps27l* KO model, that unlike in vitro cell culture studies which showed that almost all known MDM2-binding ribosomal proteins acts as p53 activators upon overexpression (*Zhang and Lu, 2009*; *Zhou et al., 2012*), Rps27l actually activates p53 upon disruption, given the fact that (a) *Rps27l* deletion caused p53 accumulation in multiple organs as well as in primary MEFs, and (b) all *Rps27l*-null phenotypes, such as depletion of HPSCs and bone marrow, growth retardation and postnatal death, can be fully rescued by *Trp53* deletion. It is conceivable that the discrepancy between in vitro and in vivo studies is likely attributable to the fact that most cell culture studies were conducted (a) in human cancer cell lines with multiple genetic alterations, (b) under artificial overexpression and/or (c) partial RNA silencing conditions. Nevertheless, through this knockout study we conclusively showed that *Rps27l* disruption induces p53 under the physiological conditions in a manner independent of DNA damage. It is also interesting in our finding that consistent with enhanced binding of Mdm2-Rpl11, but not Mdm2-Rpl5, p53 induction triggered by *Rps27l* disruption is dependent of Rpl11, but independent of Rpl5, although both are key ribosomal proteins, released from nucleolus upon ribosomal stress, to inhibit Mdm2-mediated p53 degradation (*Golomb et al., 2014*). Furthermore, we made a novel observation that *Rps27l* disruption caused an increased binding of Mdm2 selectively to Rpl11, Rpl23, Rps7, Rps14, and Rps27, but not to Rpl5, Rps27a, and Rps19. Whether these RPs with enhanced Mdm2 binding indeed contribute to p53 activation in *Rps27l-deficient* MEFs is an interesting topic warranting future investigation.

## Rps27l regulates the Mdm2-Mdm4 E3 ligase activity towards p53

Our *Rps27l* acute depletion experiment conducted in both *Trp53*-wt and *Trp53*-null MEFs clearly showed a sequential change in the levels of Mdm2 (increase), Mdm4 (decrease) and p53 (increase), indicating that (1) *Rps27l* depletion induces p53-independent Mdm2 increase and (2) increased Mdm2 fails to degrade p53. Our follow-up experiments mechanistically addressed these two fundamental

questions: how Mdm2 is increased upon *Rps27l* disruption and why increased Mdm2 fails to degrade p53 in the absence of Rps27l. For first question, we found that while Rps27l has no direct effect on Mdm2 mRNA transcription, it indeed negatively regulates the Mdm2 protein stability with Mdm2 protein half-life being extended or shortened upon *Rps27l* depletion or overexpression, respectively. It is conceivable that in wt cells under unstressed conditions, Rps27l binds to Mdm2 on its acidic domain (*Xiong et al., 2011*), which has potential to block the binding of other ribosomal proteins (e.g., L11/S7/S27a/S14) to Mdm2 on the same domain (internal competition). Upon *Rps27l* disruption, which triggers ribosomal stress, other ribosomal proteins are released from nucleoli and bind to Mdm2 on an unoccupied acidic domain to inhibit Mdm2 self-ubiquitylation, leading to its stabilization. To address the second question, we showed that the total Mdm4 level was significantly lower in *Rps27l$^{-/-}$* than in *Rps27l$^{+/+}$* MEFs. We also showed by both loss-of-function and rescue-of-function studies that Mdm2-mediated Mdm4 ubiquitylation and degradation is negatively regulated by Rps27l, being enhanced upon *Rps27l* depletion and reduced upon RPS27L overexpression. Thus, enhanced Mdm2-RPs binding (such as Mdm2-Rpl11 binding) as a result of ribosomal stress in response to *Rps27l* disruption would facilitate Mdm2-dependent Mdm4 degradation, which is consistent with a previous observation (*Gilkes et al., 2006*). Furthermore, we found that although Mdm2-Mdm4 has a better affinity upon *Rps27l* deletion, a much reduced amount of Mdm4 in *Rps27l$^{-/-}$* MEFs results in a reduced level of Mdm2-bound Mdm4, thus lesser formation of the Mdm2-Mdm4 heterodimer. Given that Mdm2-Mdm4 heterodimer is the most stable and active form of E3 ligase for targeted p53 degradation, as shown in both in vitro (*Kawai et al., 2007*; *Wang et al., 2011*) and in vivo (*Francoz et al., 2006*; *Garcia et al., 2011*; *Huang et al., 2011*; *Pant et al., 2011*) studies, compromised Mdm2-Mdm4 ligase activity in *Rps27l$^{-/-}$* MEFs leads to a reduced p53 ubiquitylation and degradation, consequently an extended p53 protein half-life and increased p53 levels.

## Lessons from the *Rps27l* mutant mice

While numerous gene knockout studies have shown an involvement of p53 during embryogenesis, the causal role played by p53 was best demonstrated by the total knockout of Mdm2 (*Jones et al., 1995*; *Montes de Oca Luna et al., 1995*) or Mdm4 (*Parant et al., 2001*; *Finch et al., 2002*; *Migliorini et al., 2002*) and by the knock-in (KI) of mutant Mdm2$^{C462A}$ (*Itahana et al., 2007*), Mdm4$^{C462A}$ (*Huang et al., 2011*), or Mdm4$^{\Delta RING}$ (*Pant et al., 2011*). In all cases, p53 is accumulated to robust levels to kill the embryos at the early stage of embryogenesis, which can only be rescued by homozygous, but not heterozygous, deletion of *Trp53* (*Jones et al., 1995*; *Montes de Oca Luna et al., 1995*; *Parant et al., 2001*; *Itahana et al., 2007*; *Huang et al., 2011*; *Pant et al., 2011*). In addition, the embryonic lethality of *Mdm2$^{-/-}$* mice can also be rescued by *Trp53$^{515C}$*, a hypomorphic allele of *p53*, which encodes p53$^{R172P}$ (*Abbas et al., 2010*). While *Mdm2$^{-/-}$;Trp53$^{515C/515C}$* mice also die by postnatal day 13, the death is mainly attributable to depletion of HPSCs in postnatal bone marrows, but not in fetal liver due to p53$^{R172P}$-induced ROS generation, and subsequent senescence and cell death (*Abbas et al., 2010*). In contrast, *Rps27l* disruption causes a moderate increase of p53 which is insufficient to induce embryonic lethality, but sufficient to induce postnatal lethality, resulting from apoptotic depletion of HPSCs in both fetal liver and neonatal bone marrow. The *Trp53* dosage effect has also been seen in *Mdm2$^{+/-}$* and *Mdm4$^{+/-}$* double heterozygous embryos and mice (*Terzian et al., 2007*).

Besides its engagement in the Mdm2/Mdm4 KO/KI studies, p53 was found to be activated upon impairment of ribosome biogenesis in several RP-deficient mouse models (*Bursac et al., 2014*). For example, liver specific deletion of *Rps6* prevented hepatocytes from re-entering the cell cycle after partial hepatectomy due to increased p53, which can be rescued by *Trp53* deletion (*Volarevic et al., 2000*; *Fumagalli et al., 2009*). Developmental defects derived from tissue specific deletion of *Rps6* in T cells or oocytes was largely due to p53 activation, and rescued by *Trp53* deletion (*Sulic et al., 2005*; *Panic et al., 2006*). Likewise, p53 activation was involved in defective αβ T cell development in *Rpl22* knockout mice and this deficiency was completely rescued by *Trp53* deletion (*Anderson et al., 2007*). p53 activation was also involved in congenital malformations in *Rpl24*-deficient mice which were largely rescued by p53 ablation (*Barkic et al., 2009*). Furthermore, phenotypes such as cerebellar ataxia, pancytopenia and epidermal hyperpigmentation seen in *Rpl27a* mutant mice was rescued in a haploinsufficient *Trp53* background (*Terzian et al., 2011*), whereas the dark skin as a result of p53-dependent epidermal melanocytosis seen in Rps6, Rps19, or Rps20 mutant mice was also rescued by *Trp53* deletion (*McGowan et al., 2008*). Finally, p53 appears to be causally related to the defects seen in the Treacher Collins syndrome (TCS), a congenital disorder of craniofacial development arising

from mutations of TCOF1, which encodes the nucleolar phosphoprotein Treacle and whose haploin-sufficiency perturbs mature ribosome biogenesis to trigger p53 activation (*Jones et al., 2008*).

Only one mouse KO model involving the deletion of an Mdm2-binding ribosomal protein Rps14, along with other seven known genes (deletion of Cd74-*Nid67* interval) in the 5q-syndrome, was previously reported (*Barlow et al., 2010*). Hematological abnormalities associated with increased apoptosis in bone marrow progenitor cells can be rescued only by homozygous *Trp53* deletion (*Barlow et al., 2010*). To our best knowledge, our study is the first targeted inactivation of a single Mdm2-binding ribosomal protein in mouse and demonstrated that (1) Rps27l is required in vivo to keep p53 in check and (2) developmental defects and postnatal death upon *Rps27l* disruption are rescued by the deletion of single *Trp53* allele, indicating that moderate increase of p53 is sufficient to tip the life-death balance to the death. It is worth noting that postnatal death upon *Rps27l* disruption occurs under the wt background of its family member, *Rps27* as well as other ribosomal genes encoding all known Mdm2-binding ribosomal proteins. Thus, for hematopoiesis, Rps27l plays a non-redundant role in reducing p53 levels via stabilizing the Mdm2-Mdm4 complex.

### Rps27l regulates genomic stability

The viability of $Rps27l^{-/-};Trp53^{+/-}$ mice provided us an opportunity to study the role of Rps27l in spontaneous tumorigenesis. Surprisingly, while moderate p53 increase is expected to decrease the tumor incidence as shown in *Mdm2* hypomorphic mice (*Mendrysa et al., 2006*) and in $Mdm2^{+/-}$ or $Mdm4^{+/-}$ mice (*Alt et al., 2003*; *Terzian et al., 2007*), our $Rps27l^{-/-};Trp53^{+/-}$ mice have a much higher incidence than $Rps27l^{+/+};Trp53^{+/-}$ mice to develop spontaneous T-lymphoblastic and in rare case B-cell lymphomas, resulting in a shortened life-span. In 97% of tumor tissues genotyped, wt *Trp53* allele was deleted, indicating that *Rps27l* depletion imposes a selection pressure against p53. Furthermore, the genome of these lymphoma cells is highly unstable with a strike aneuploidy rate of ~63%, which is remarkably higher than those derived from *Trp53*-null mice with an aneuploidy rate of 30–35% (*Kibe et al., 2012*). Even in diploid population, abnormal centromere–centromere fusions were found. Thus, our study provides an in vivo demonstration that Rps27l is required for the maintenance of genomic stability, which is consistent with a previous report, showing that RPS27L knockdown in HCT116 colon cancer cells may trigger genomic instability (*Li et al., 2007*).

One more striking observation we made in this study is that the genomic instability occurs in early passage primary MEFs with the genotype of $Rps27l^{-/-};Trp53^{+/-}$, but not of $Rps27l^{-/-};Trp53^{+/+}$. In other word, Rps27l plays a limited role in the maintenance of genomic stability, if p53 is normal. However, when one allele of *Trp53* is deleted or mutated (such as in the case of Li-Fraumeni syndrome or at various stage of tumorigenesis), Rps27l becomes critically essential in preventing the loss of p53 heterozygosity. Thus, its loss or decreased expression by any means would exacerbate genomic instability and provide selection pressure against p53, eventually leading to tumor development. Potential tumor suppressive function of RPS27L is further supported by an observation that high level of RPS27L expression predicted a better prognosis in colon cancer patients (*Huang et al., 2013*).

In summary, our results provide strong pieces of in vivo evidence that Rps27l precisely regulates p53 threshold. In normal cells with wt p53 ($Trp53^{+/+}$ status), Rps27l appears to be a physiological inhibitor of p53 through optimizing Mdm2-Mdm4 E3 ligase activity towards p53 to keep p53 in check. *Rps27l* disruption triggers ribosomal stress to increase Mdm2 for targeted Mdm4 degradation, leading to p53 activation (in an Rpl11 dependent manner) and subsequent apoptosis induction and postnatal death. Under $Trp53^{+/-}$ status, Rps27l acts as a tumor suppressor by maintaining the genomic stability and preventing *Trp53* deletion. *Rps27l* disruption triggers genomic instability and confers selection pressure for p53 inactivation, leading to lymphamagenesis (*Figure 9*). Whether the RPS27L level determines the early-onset of human cancers in Li-Fraumeni syndrome patients with a germ-line $TP53^{+/-}$ status is certainly an intriguing question that deserves further investigation.

## Materials and methods

### Generation of Rps27l gene trapped mice

Germline-transmitted heterozygous *Rps27l* mice generated from an ES cell clone (IST11658B7, C57BL/6) were obtained from the Texas A&M Institute for Genomic Medicine (TIGM, College Station, TX). Germline transmission was confirmed by PCR and Southern blotting. Mice bearing the *Trp53*-null allele (deletion of exons 2–7) (*Jacks et al., 1994*) were provided by Dr Yuan Zhu (University of Michigan,

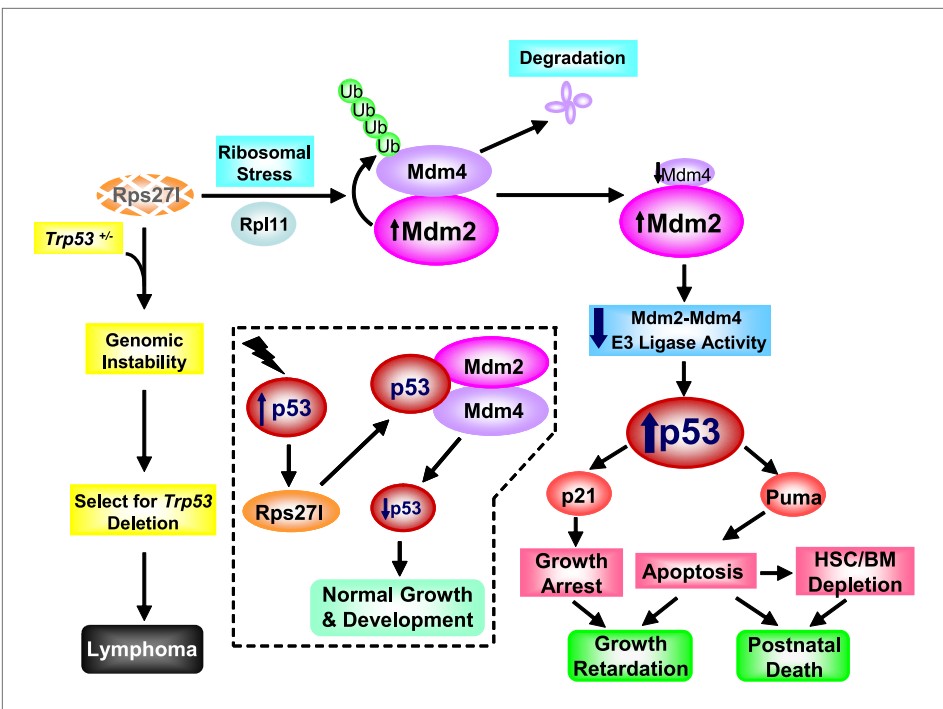

**Figure 9**. Rps27l is a p53 regulator and also a p53 'goalkeeper'—a working model. In normal cell with wild-type p53 (*Trp53*$^{+/+}$), Rps27l, upon induction by p53 in response to various stresses, stabilizes the Mdm2-Mdm4 heterodimer to form an optimal E3 ligase complex for effective p53 ubiquitylation and degradation, thus keeping p53 level in check, leading to normal growth and development (dotted area). *Rps27l* disruption causes an imbalance in ribosomal protein levels and triggers ribosomal stress to stabilize Mdm2. Increased Mdm2 adapts a conformation that favors the Mdm4 ubiquitylation, leading to a reduced Mdm2-Mdm4 complex and compromised p53 ubiquitylation with an ultimate increase in p53. Moderately increased p53 is sufficient to induce growth retardation and apoptosis by transactivating p21 and Puma, respectively, leading to the depletion of HSPCs and bone marrow, and eventually postnatal death. In cancer-prone cells with a *Trp53*$^{+/-}$ status, Rps27l plays an essential role in keeping genome integrity. *Rps27l* disruption triggers genomic instability, followed by selection for *Trp53* deletion. The cells with *Trp53* deleted are selected for and outgrow to form spontaneous lymphoma.

Ann Arbor, MI). The genetic background of *Rps27l* mutant mice is pure C57/BL6, while *Rps27l/ Trp53* mice are the hybrids of 129 Svj and C57BL/6. C57BL/6 Ly5.2 (CD45.1) mice were purchased from the National Cancer Institute. All procedures were approved by the University of Michigan Committee on Use and Care of Animals. Animal care was provided in accordance with the principles and procedures outlined in the National Research Council Guide for the Care and Use of Laboratory Animals.

## PCR-based genotyping
Genomic DNA was isolated from mouse tail tips according to University of Michigan Transgenic Animal Model Core protocol (http://www.med.umich.edu/tamc/tDNA.html). Mice were genotyped using the primer set of LTR-Rev 2: 5′-ACC TGA AAT GAC CCT GTG CCT TA-3′ and IST11658B7-f: 5′-TTG ATG GCT ACC CAG CCA AAC G-3′ for *Rps27l* mutant (325 bp) and IST11658B7-f and IST11658B7-r2: 5′-ACG TAT CCT TTA CCT GGC TCC C-3′ for *Rps27l* wt (525 bp). The primer sets for *Trp53* genotyping are p53x6.5: 5′-ACA GCG TGG TGG TAC CTT AT-3′; p53x7: 5′-TAT ACT CAG AGC CGG CCT-3′; and p53 Neo19: 5′-CAT TCA GGA CAT AGC GTT-3′.

## Southern analysis
Genomic DNA was isolated from mouse tails and digested with *EcoRI* or *PstI*. The 5′ end probe (644 bp) was generated by PCR using the primer set, S27LKO-5pb-F: 5′-TAA GCC AGG GGG TCA ATA-3′ and S27LKO-5pb-R: 5′-CTC CCC TGT TCA TTG TGC-3′. The β-Gal probe (686 bp) was generated by PCR using the primer set, S27LKO-BG-F: 5′-GGC GTA ATA GCG AAG AGG-3′, and S27LKO-BG-R:

5'-TTC ACC CTG CCA TAA AGA-3'. All the probes were confirmed by DNA sequencing. Southern blot analysis was carried out as described (*Tan et al., 2009*).

## Generation and maintenance of MEFs

MEF cells were generated from day E13.5 embryos with indicated genotypes as described (*Tan et al., 2009*), and cultured in DMEM with 15% FBS, 2 mM L-Glutamine, 0.1 mM MEM non-essential amino acids at 37°C in a 5% CO2 humidified chamber.

## Immunofluorescence

Paraffin sections were deparaffinized, rehydrated, and analyzed by immunofluorescence (*Wang et al., 2009a*). The tissue sections were incubated with antibody against cleaved caspase-3 (Cell signaling Technology, Danvers, MA) in blocking solution overnight. MEF cells were left untreated or treated for 24 hr with ribosomal stress inducer, actinomycin D (5 nM), followed by immune-staining with antibody against B23 (Sigma, St. Louis, MO). The secondary antibodies were conjugates of Alexa Fluor 488 or Alexa Fluor 594 (Life Technologies, Grand Island, NY). DAPI (Life Technologies) was used as nuclear counterstaining. Stained tissues or cells were examined under a fluorescence microscope (Olympus, Center Valley, PA).

## Western blotting and immunoprecipitation

Cells or tissues were harvested, lysed and subjected to Western blotting or immunoprecipitation (*Macias et al., 2010*), using various antibodies as follows: RPS27L or RPS27 polyclonal rabbit antibody was raised and purified as described (*He and Sun, 2007*), p53 (1C12 from Cell signaling technology and CM5p from Leica Microsystems, Buffalo Grove, IL), Mdm2 4B2 and Mdm4 7A8 (gifts from Dr Jiandong Chen), rabbit polyclonal Mdm4 (gift from Dr Aart Jochemsen, used for IP), Rpl5, Rpl11, and Rpl23 (gifts from Drs Yanping Zhang and Hua Lu), Rps7 (gift from Dr Ruiwen Zhang), Rps27a (gift from Dr Mushui Dai), Rps14 and Rps19 (Santa Cruz Biotechnology, Santa Cruz, CA), rabbit polyclonal Ab for mouse Mdm2 (raised against mouse Mdm2 peptide EQTPLSQESDDYSQPSTSSS, made by Yenzym, San Francisco, CA, used for IP), human MDM2 (Ab-1, Calbiochem, San Diego, CA), p21 (BD Biosciences, San Jose, CA), and β-actin (Sigma).

## Flow cytometry

For surface staining, cells were incubated with the indicated antibodies in staining buffer (HBSS with 2% FBS) for 20 min at 4°C. The following antibodies were purchased from eBioscience, San Diego, CA: Fluorescein isothiocyanate (FITC)-conjugated anti-CD3e (clone 145-2C11), anti-CD4 (GK1.5), anti-CD8a (53–6.7), anti-CD16/CD32 (93), anti-CD43 (eBioR2-60), anti-CD45.1 (A20), and anti-CD45.2 (104); phycoerythrin (PE)-conjugated anti-CD3e (M1/69), anti-B220 (RA3-6B2), anti-Mac-1 (M1/70), anti-Gr-1 (RB6-8C5), anti-Ter119 (TER-119), and anti-CD45.2 (104); PE-Cy7–conjugated anti-CD4 (RM4-5), and anti-B220 (RA3-6B2); allophycocyanin (APC)-conjugated anti-CD8a (53–6.7), anti-CD45.1 (A20), anti-CD48 (HM48-1), anti-CD71 (R17217), and anti-Gr-1 (RB6-8C5); APC-eFluor780–conjugated anti-CD117 (c-Kit, clone 2B8); and eFluor660-conjugated anti-CD34 (RAM34). The following antibodies were purchased from BD Biosciences: PerCP-Cy5.5–conjugated anti-B220 (RA3-6B2), and anti-Mac-1 (M1/70); PE-Cy7–conjugated anti-CD45.2 (104), and anti-Ly-6A/E (Sca-1, clone D7). Lineage markers included B220 (B cells), CD3 (T cells), Gr-1 (granulocyte), Mac-1 (myeloid cells), and Ter119 (erythrocytes). PerCP-Cy5.5-conjugated anti-CD150 Ab (clone TC15-12F12.2) was purchased from BioLegend, San Diego, CA, and FITC-conjugated Annexin V and FITC-conjugated anti-BrdU antibody were from BD Biosciences. All FACS analyses were performed on an LSR II flow cytometer (BD Biosciences), and data were analyzed with FlowJo software (*Tang et al., 2012*).

## Transplantation assays

6–8-week-old C57BL/6 Ly5.2 (CD45.1) recipient mice were lethally irradiated with a [137]Cs source delivering 170 rad per min for a total dose of 1100 rads. Given numbers of fetal liver cells from E14.5 embryos (CD45.2) were either injected alone or mixed with recipient-type (CD45.1) competitive BM cells. The cells were injected into recipients through the tail vein within 24 hr after irradiation. Reconstitutions were measured by flow cytometry of peripheral blood at the time points indicated (*Chen et al., 2009*; *Tang et al., 2012*).

## Immunohistochemistry

Immunohistochemical staining was performed on the DAKO Autostainer (DAKO, Carpinteria, CA) using diaminobenzadine (DAB) as the chromogen. After dewaxing and rehydration, serial sections

were labeled with p53 (CM5p, Leica Microsystems), after 10 mM citrate buffer, pH6 microwave epitope retrieval. LSAB+ (DAKO) was employed as the detection system. Appropriate negative (no primary Ab) and positive controls were stained in parallel.

## In vitro ubiquitylation assay

MEF cells treated with MG132 for 4 hr were lysed and IP with Mdm2 Ab. The Mdm2/Mdm4-p53 complex were then incubated with 10 μg ubiquitin, 375 ng UBE1, 150 ng Ubc5Hb (Boston Biochem, Cambridge, MA), and 30 μl reaction buffer (50 mM Tris pH7.5, 2.5 mM MgCl$_2$, 15 mM KCl, 1 mM DTT, 0.01% Triton-X-100, 1% glycerol) in the presence of 4 mM ATP. The mixture was incubated at 37°C for 60 min with continuous vortexing, and subjected to IB after boiling in SDS sample buffer (*Cheng et al., 2009*).

## Ribosomal profiling

The profiling was conducted as described (*Zhu et al., 2012*) with modifications. Briefly, MEFs were treated with cycloheximide (100 μg/ml) in growth medium for 30 min at 37°C. Cells were washed with PBS containing cycloheximide and then lysed in extraction buffer containing 20 mM HEPES (pH 7.5), 5 mM MgCl$_2$, 150 mM KCl, 100 μg/ml cycloheximide, 1 mM DTT, 0.5% Triton-X 100, 0.5% Sodium deoxycholate, and RNase inhibitor. Extracts were spun for 10 s to pellet nuclei and then cleared by centrifugation at 10,000×*g* for 10 min at 4°C. The cytoplasmic extracts were loaded onto 4.5 ml sucrose gradients (10%–50%) buffered in 20 mM HEPES (pH 7.5), 100 mM KCl, 5 mM MgCl$_2$, 1 mM DTT. Gradients were subjected to ultra-centrifugation using a Beckman SW50.1 Rotor at 40,000 rpm for 100 min at 4°C. Gradients were then fractionated measured by optical density at 254 nm.

## rRNA processing

MEF cells on 60 mm dish were starved of methionine in methionine-free medium for 30 min and then pulse-labeled for 30 min in 1 ml medium containing 50 μCi of L-[*methyl*-$^3$H]-methionine (MP Biochemicals, Santa Ana, CA). After rinsing with 10 × methonine (0.3 mg/ml) medium, cells were incubated in 10 × methonine medium for indicated time periods. Total RNA was isolated using RNeasy kit (Qiagen, Valencia, CA) and subjected to 1% Agarose formaldehyde gel electrophoresis and then transferred on Nylon membrane. The membrane was dried and sprayed with EN3HANCE (PerkinElmer, Waltham, MA) and then exposed at −80°C for a week (*Itahana et al., 2003*).

## In vivo ubiquitylation assay

Human lung H1299 cells were transiently transfected with various plasmids. Cells were harvested 24 hr post transfection after last 4 hr MG132 treatment and split into two aliquots with one for direct Western blotting analysis and the other for in vivo ubiquitylation assay as described (*Gu et al., 2007*). Briefly, cell pellets were lysed and incubated with Ni-NTA beads (Qiagen) at room temperature for 4 hr. Beads were then washed and incubated with elution buffer at room temperature for 20 min. The eluted proteins were analyzed by Western blotting.

## Lentivirus-based shRNA

The sequences of scrambled control siRNA and RPS27L siRNA have been described (*Xiong et al., 2011*). Short hairpins targeting mouse Rpl11 (targeting sequence: 5′-CGG GAG TAT GAG TTG CGG AAA-3′) or Rpl5 (targeting sequence: 5′-CCC TCA TAG TAC CAA ACG ATT-3′) was cloned into pLKO.1-puro vector. Two additional Rpl5-silencing clones (TRCN0000104427 and TRCN0000104429) were purchased from Thermo Fisher Scientific, Waltham, MA. Lentiviral particles were produced by University of Michigan Vector Core.

## Metaphase preparation

Primary thymic lymphoma cells from *Rps27l*$^{-/-}$;*Trp53*$^{+/-}$ mice or primary MEFs at early passage were treated with 0.2 μg/ml KaryoMAX colcemid solution (Life Technologies) for 2 hr before harvesting for metaphase preparation. Metaphase spreads were stained by incubation in 4% KaryoMAX Giemsa solution (Life Technologies) for 15 min, followed by observation under a light microscope. Chromosome numbers were counted (*Wu et al., 2011*).

## Real-time PCR

Total RNA was isolated from MEF cells with Trizol reagent (Life Technologies). Complementary DNA was made from RNA with Superscript III (Life Technologies). Real-time PCR was performed on a 7500 Real

Time PCR system (Life Technologies). The cycling program was set as follows: 50°C 2 min, 95°C 10 min for the PCR initial activation and 45 cycles of denaturation at 95°C for 15 s, annealing and extension at 60°C for 1 min. The sequences of p53, Mdm2, p21, Mdm4, 45S rRNA, and GAPDH are as follows: p53-RT-F: 5′-GAG AGT ATT TCA CCC TCA AGA TCC G-3′, p53-RT-R: 5′-CCC CAC TTT CTT GAC CAT TGT TT-3′; Mdm2-RT-F: 5′-GAT GAG GAT GAT GAG GTC TAT CGG-3′, Mdm2-RT-R: 5′-TCT GGA AGC CAG TTC TCA CGA A-3′; p21-RT-F: 5′-ACT TCC TCT GCC CTG CTG CA-3′, p21-RT-R: 5′-CGC TTG GAG TGA TAG AAA TCT GTC A-3′; Mdm4-RT-F: 5′-TTT ACA GAC AAA TCA GGA TAT AGG TA-3′, Mdm4-RT-R: 5′-GTA CAC TGC CAC TCA TCC TCA-3′; 45S-F: 5′-ACA CGC TGT CCT TTC CCT ATT AAC ACT AAA-3′, 45S-R: 5′-AGT AAA AAG AAT AGG CTG GAC AAG CAA AAC-3′; GAPDH-F: 5′-GCC GCC TGG AGA AAC CTG CC-3′, GAPDH-R: 5′-GGT GGA AGA GTG GGA GTT GC-3′ (*Poortinga et al., 2004*).

For quantitative measurement of *Trp53* dosage, genomic DNA were isolated from MEFs and amplified with the following primer sets: p53-del-F2: 5′-CCT GAT CGT TAC TCG GCT TGT C-3′, p53-del-R2: 5′-CAA CTG CAC AGG GCA CGT CT-3′; p53-del-F4: 5′-GGC TTC TGA CTT ATT CTT GCT CTT A-3′, p53-del-R4: 5′-AGA CCT CGG GTG GCT CAT AA-3′; p53-del-F5: 5′-GAG GTA GGG AGC GAC TTC ACC-3′, p53-del-R5: 5′-GGT AAG GAT AGG TCG CGG GTT-3′; GAPDH-F: 5′-GTA TGA CTC CAC TCA CGG CAA A-3′, GAPDH-R: 5′-GGT CTC GCT CCT GGA AGA TG-3′.

## [$^{35}$S]-methionine metabolic labeling

For determination of p53 protein synthesis, MEF cells were treated for 1 hr in methionine-free and cysteine-free DMEM, containing 10% dialyzed FCS and 50 µM MG132. Cells were then labeled with 250 µCi/ml of [$^{35}$S]-methionine (MP Biochemicals), followed by immunoprecipitation with anti-p53 antibody (1C12). Immunoprecipitates, along with whole cell extract, were then subjected to SDS-PAGE and autoradiography (*Wang et al., 2009b*).

For determination of global nascent protein synthesis, MEF cells on 60 mm dish were starved in methionine-free and cysteine-free DMEM with dialyzed FCS for 30 min and then pulse-labeled with 1 ml [$^{35}$S]-methionine (200 µCi/ml) for 1 hr. After PBS washing, cells were lysed in 0.5% NP-40 buffer. Equal amounts of protein extracts were precipitated with 20% ice-cold TCA for 30 min on ice, and dissolved in 0.1 N NaOH. Total [$^{35}$S]-methionine incorporation was measured with a liquid scintillation counter and plotted as counts per min/µg protein. Equal amounts of protein were also subjected to SDS-PAGE and autoradiography (*Lindstrom and Zhang, 2008*).

## *Trp53* exons amplification

To determine the deletion of *Trp53* wt allele, genomice DNA were isolated and amplified with the following primer sets: Exon-1-F: 5′-ATC GGT TTC CAC CCA TTT TG-3′, Exon-1-R: 5′-ATA CAC TCC CGT TCA TCC CG-3′; Exon-2-F: 5′-TAC CTC TGC TCA GCC CCC G-3′, Exon-2-R: 5′-TTA CAG ACA CCC AAC ACC ATA CCA-3′; Exon-3-F: 5′-GCA GGG TCT CAG AAG TTT GAG G-3′, Exon-3-R: 5′-GTG GAT GGG ACA AAG AAG AAC C-3′; Exon-4-F: 5′-TTG GGC TTT GGT GTT GGG-3′, Exon-4-R: 5′-AGG CTG AAG AGG AAC CCC C-3′; Exon-5-F: 5′-CGG GGA GTT GTC TTT CGT GT-3′, Exon-5-R: 5′-TAA GAG CAA GAA TAA GTC AGA AGC C-3′; Exon-6-F: GTA AGC CCT CAA CAC CGC C-3′, Exon-6-R: 5′-GAC TCA GCG TCT CTA TTT CCC G-3′; Exon-7-F: 5′-TCC AGC AGG TGT GCC GAA-3′, Exon-7-R: 5′-AAC CCC GAG AAG CCA CAG A-3′; Exon-8-F: 5′-TCT GTG GCT CTC GGG GT TC-3′, Exon-8-R: 5′-GGA AGG AGA GAG CAA GAG GTG AC-3′; Exon-9-F: 5′-CGG AGG AGC CTG TTG AGC TT-3′, Exon-9-R: 5′-CAG CCT CAG AGC ATG AGC TC-3′; Exon-10-11-F: GAG CCA GCT TAA GTT GGG AAC-3′, Exon-10-11-R: 5′-GAC AGC AAG AGA GGG GGG-3′.

## Statistical analysis

The two-tailed Student's *t* test was used for the comparison of parameters between groups. Survival analysis was performed by Kaplan–Meier analysis. Statistical significance was determined as $p < 0.05$.

## Acknowledgements

We would like to thank Drs Jiandong Chen, Aart G Jochemsen, Yanping Zhang, Hua Lu, Ruiwen Zhang, and Mushui Dai for providing antibodies; Dr Janine Maddock for her help in setting up ribosomal profiling experiment. We also thank Dr Yuan Zhu for providing us *Trp53*-deficient mice and discussions. This work is supported by the NCI grants (CA118762, CA156744, CA170995 and CA171277) to YS and Susan G Komen for the Cure PDF Grant (PDF12230424) to YZ.

## Additional information

### Funding

| Funder | Grant reference number | Author |
| --- | --- | --- |
| National Cancer Institute | R01 CA118762 | Xiufang Xiong, Yongchao Zhao, Yi Sun |
| National Cancer Institute | R01 CA171277 | Xiufang Xiong, Yongchao Zhao, Yi Sun |

The funder had no role in study design, data collection and interpretation, or the decision to submit the work for publication.

### Author contributions

XX, YZ, Conception and design, Acquisition of data, Analysis and interpretation of data; FT, DW, DT, XW, Acquisition of data, Analysis and interpretation of data; YL, PZ, Conception and design, Analysis and interpretation of data; YS, Conception and design, Analysis and interpretation of data, Drafting or revising the article

### Ethics

Animal experimentation: All the animal procedures were approved by the University of Michigan Committee on Use and Care of Animals (Protocol # PRO00004764). Animal care was provided in accordance with the principles and procedures outlined in the National Research Council Guide for the Care and Use of Laboratory Animals.

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
