## [Decision Letter]

Thank you for sending your work entitled “Ribosomal protein S27-like is a physiological inhibitor of p53 that paradoxically suppresses tumorigenesis” for consideration at *eLife*. Your article has been favorably evaluated by a Senior editor, a Reviewing editor, and 2 reviewers.

The Reviewing editor and the reviewers discussed their comments before we reached this decision, and the Reviewing editor has assembled the following comments to help you prepare a revised submission.

There are novel and potentially important findings reported in this manuscript that make it appropriate in principle for publication in *eLife.* The authors convincingly show that the defects seen in their Rps27L knockout mouse are p53 dependent and apoptotically related. They also nicely highlight the roles of Rps27L and p53 in maintaining HSPC's and peripheral blood cells. Especially interesting is the observation that Rps27L-deficiency accelerates the formation of spontaneous malignant tumors in p53-heterozygous mice. Nevertheless the referees feel that there are some deficiencies that need to be addressed.

1) There is ample evidence in the literature that the source of the p53 stimulation upon depletion of specific ribosomal proteins is aberrant ribosome assembly. It is essential to determine whether Rps27L-deficiency impairs ribosome biogenesis (rDNA transcription, rRNA processing etc.). Furthermore, recent studies have documented the central role of RPL5 and RPL11 in this p53-dependent signaling pathway. Is activation of p53 by Rps27L-deficiency dependent on RPL5 and RPL11? Moreover, it would be important to test if the observed displacement of NPM from the nucleolus to the nucleoplasm in Rps27L-deficient MEFs is a consequence of p53 activation.

2) The finding that Rps27L-deficiency in MEFs does not influence the polysome profiles, suggests that Rps27L does not have a role in ribosome formation. But the quality of the polysome profiles (Figure 5 and Figure S5A) is not satisfactory. This limits the conclusions that can be drawn from such experiments. Is it possible that the source of the p53 stimulation is aberrant protein synthesis, which might be due to the missing Rps27L?

3) Presented evidence clearly demonstrates that Rps27L-deficiency up-regulates p53 in vivo and that there is a selective pressure against p53 in *Rps27l*^*-/-*^;p53^*+/-*^ mice. The authors suggest that Rps27L depletion is responsible for genomic instability by a mechanism that is independent of p53 loss. The only results that support that possibility are shown in Figure 7 and Figure 7. The acquisition of genomic instability in early passage *Rps27l*^*-/-*^;p53^*+/-*^ MEFs (prior to the loss of the second p53 allele) could be due to a selective pressure against p53 upstream regulators (for example ARF). It is critical to assess the p53 protein levels and p53 activity in these MEFs before a direct link (independent of p53 loss) between Rps27L depletion and genomic instability can be made.

4) Although Figure 4 shows that Mdm2-bound Mdm4 is reduced in *Rps27l*^*-/-*^ cells, this result does not clearly support the notion that reduced Mdm2/Mdm4 heterodimer formation in *Rps27l*^*-/-*^ mice is solely due to decreased Mdm4 levels. The loss of Rps27L appears to also somehow affect the binding affinity between Mdm2 and Mdm4. The protein levels in the whole cell extract show that Mdm4 levels are hardly different in the *Rps27l*^*-/-*^ MEFs after the MG132 treatment, and there is visibly more Mdm2 pulled down in the *Rps27l*^*-/-*^ extracts, with significantly less Mdm4 associated with it. For the author's model to be cohesive this issue needs to be convincingly addressed.

5) While the authors demonstrate that Rps27L is physiologically relevant in p53 regulation, defining Rps27L as an inhibitor of p53 simply because its depletion induces p53 may not be valid. As the authors note, the depletion of Rps27L does appear to cause ribosomal stress, and ribosomal stress causes p53 activation, but that does not directly prove that its biological function is to inhibit p53 activation. Does the overexpression of Rps27L in MEFs suppress WT levels of p53? Does Rps27L normally prohibit other ribosomal proteins from stabilizing Mdm2? A logical conclusion from this study is that in the presence of normal levels of Rps27L there is no cellular stress (defective ribosome biogenesis or protein synthesis?) that triggers the p53 response.

[Editors' note: further clarifications were requested prior to acceptance, as described below.]

Thank you for resubmitting your work entitled “Ribosomal protein S27-like is a physiological regulator of p53 that suppresses genomic instability and tumorigenesis” for further consideration at *eLife*. Your revised article has been favorably evaluated by Tony Hunter (Senior editor), a member of the Board of Reviewing Editors, and the original two reviewers.

While one referee indicated that he/she was satisfied with all your revisions, the other reviewer still has some comments to be addressed as follows:

1) The authors argue that Rps27l depletion doesn't affect rRNA processing. However, the results shown in Figure S5 C demonstrate that the amount of labeled 18S rRNA was reduced in *Rps27l*^*-/-*^ cells. It will be necessary to quantify these results. The original Conclusion apparently needs to be modified.

2) It will be necessary to increase the size of the immunofluorescence images in Figure 5 to better appreciate the redistribution of B23 protein form the nucleolus to the nucleus.

3) The results shown in Figure 5 clearly show that Rpl11 is required for p53 activation upon Rps27l deficiency. It would be important to include Rpl11 in the model (Figure 9).

4) Based on available literature it is completely unexpected that depletion of Rpl5 in wt cells upregulates the p53 protein levels (Figure 5). It will be necessary to validate this result with multiple effective siRNAs against Rpl5.

5) The authors argue that in addition to RPL11, the binding of Rpl23, Rps7, Rps14 and Rps27 to Mdm2 could be relevant for p53 activation in Rps27l-deficient MEFs. What is the effect of silencing of the above-mentioned RPs on p53 activation in these cells?

6) The authors demonstrated in the revised manuscript that p53 levels are higher in *Rps27l*^*-/-*^;p53^*+/-*^ than in *Rps27l*^*+/+*^;p53^*+/-*^ MEFs (Figure 8), arguing that Rps27l deficiency can cause genomic instability independently of p53 loss (Figure 8. Is the passage number of MEFs used in experiments shown in Figure 8 the same. This has to be clearly indicated in the legend text.

---

## [Author Response]

*1) There is ample evidence in the literature that the source of the p53 stimulation upon depletion of specific ribosomal proteins is aberrant ribosome assembly. It is essential to determine whether Rps27L-deficiency impairs ribosome biogenesis (rDNA transcription, rRNA processing etc.). Furthermore, recent studies have documented the central role of RPL5 and RPL11 in this p53-dependent signaling pathway. Is activation of p53 by Rps27L-deficiency dependent on RPL5 and RPL11? Moreover, it would be important to test if the observed displacement of NPM from the nucleolus to the nucleoplasm in Rps27L-deficient MEFs is a consequence of p53 activation*.

As shown in original Figure S4A (now Figure 5—figure supplement 1), our qRT-PCR analysis revealed a similar level of 45S pre-rRNA transcribed from rDNA between *Rps27l*^*+/+*^ and *Rps27l*^*-/-*^ MEFs. So, *Rps27l* deletion does not impair rDNA transcription.

We have now performed pulse-chase labeling of rRNA to examine the kinetics of rRNA processing in *Rps27l*^*+/+*^ and *Rps27l*^*-/-*^ MEFs and found no obvious difference, suggesting that *Rps27l* deletion has minimal, if any, effect on rRNA processing. This new data is shown in Figure 5—figure supplement 1, and described in the text.

In our original submission, we have shown that upon *Rps27l* deletion, the binding of Mdm2 with Rpl11, but not Rpl5 was increased (Figure 5). We have now shown that silencing of Rpl11, but not Rpl5 abrogated the p53 activation in *Rps27l*^*-/-*^ MEFs, suggesting activation of p53 by *Rps27l*-deficiency is dependent on Rpl11, but not Rpl5. This new data is now shown in Figure 5 and described in the text.

To determine whether displacement of NPM/B23 from the nucleolus to the nucleoplasm in *Rps27l*-deficient MEFs is a consequence of p53 activation, we used *Rps27l*^*-/-*^*;Trp53*^*-/-*^ double null MEFs and found that increased nucleolar B23 release was also seen in *Rps27l*^*-/-*^*;Trp53*^*-/-*^ MEFs, although to a lesser extent. Thus, it appears that nucleolus to nucleoplasm translocation of B23, triggered by *Rps27l* deletion, is largely independent of p53. This new data is now shown in Figure 5 and described in the text.

*2) The finding that Rps27L-deficiency in MEFs does not influence the polysome profiles, suggests that Rps27L does not have a role in ribosome formation. But the quality of the polysome profiles (*Figure 5
*and Figure S5A) is not satisfactory. This limits the conclusions that can be drawn from such experiments. Is it possible that the source of the p53 stimulation is aberrant protein synthesis, which might be due to the missing Rps27L?*

We repeated the polysome profiling experiment and obtained the better quality results, which showed a similar pattern between wt and null cells (Figure 5). Thus, *Rps27l* deletion has minimal, if any, effect in ribosome formation. To investigate whether *Rps27l* deficiency has any effect on protein synthesis, thus ruling in or ruling out the possibility that p53 stimulation seen in null cells is due to aberrant protein synthesis, we used three independent pairs of wt vs. *Rps27l*-null MEFs and measured the rate of protein synthesis by [^35^S]-methionine incorporation. The results showed that *Rps27l*^*+/+*^ and *Rps27l*^*-/-*^ MEFs have a similar rate of protein synthesis, thus excluding the possibility of aberrant protein synthesis. This new data is now shown in Figure 5—figure supplement 1 and described in the text. Based upon this result, we concluded that p53 activation upon *Rps27l* deletion is due to ribosomal stress.

*3) Presented evidence clearly demonstrates that Rps27L-deficiency up-regulates p53 in vivo and that there is a selective pressure against p53 in Rps27L*^*-/-*^*;p53*^*+/-*^
*mice. The authors suggest that Rps27L depletion is responsible for genomic instability by a mechanism that is independent of p53 loss. The only results that support that possibility are shown in*
Figure 7
*and*
Figure 7*. The acquisition of genomic instability in early passage Rps27L*^*-/-*^*;p53*^*+/-*^
*MEFs (prior to the loss of the second p53 allele) could be due to a selective pressure against p53 upstream regulators (for example ARF). It is critical to assess the p53 protein levels and p53 activity in these MEFs before a direct link (independent of p53 loss) between Rps27L depletion and genomic instability can be made*.

We agreed with reviewers’ assessment of our data and have now performed suggested experiments in three independent pairs of MEFs of *Rps27l*^*-/-*^*;Trp53*^*+/-*^ vs. the littermates of *Rps27l*^*+/+*^*;Trp53*^*+/-*^. In every case, both basal and induced (by ribosomal stress inducer of 5 nM ActD) levels of p53 and p53 downstream (p21 and Mdm2) are higher in R*ps27l*-null MEFs than in *Rps27l*-wt MEFs, whereas the p19/Arf level was lower in the former. These results demonstrated that *Rps27l* disruption induces genomic instability under *Trp53*^*+/-*^ background, which triggers p53 to balance the genome, and is indeed independent of p53 loss. On the other hand, *Rps27l* disruption also confers a selective pressure against p19/Arf, a p53 upstream regulator, although the underlying mechanism is unclear. Nevertheless, p19/Arf reduction would likely contribute to observed Mdm2 increase to degrade Mdm4, leading to p53 accumulation, which is consistent with our hypothesis. These newly generated data are now shown in Figure 8 with description incorporated in the text.

*4) Although*
Figure 4
*shows that Mdm2-bound Mdm4 is reduced in Rps27L*^*-/-*^
*cells, this result does not clearly support the notion that reduced Mdm2/Mdm4 heterodimer formation in Rps27L*^*-/-*^
*mice is solely due to decreased Mdm4 levels. The loss of Rps27L appears to also somehow affect the binding affinity between Mdm2 and Mdm4. The protein levels in the whole cell extract show that Mdm4 levels are hardly different in the Rps27L*^*-/-*^
*MEFs after the MG132 treatment, and there is visibly more Mdm2 pulled down in the Rps27L*^*-/-*^
*extracts, with significantly less Mdm4 associated with it. For the author's model to be cohesive this issue needs to be convincingly addressed*.

The reviewers made an excellent point. Indeed, the data shown in original Figure 4 cannot tell whether decreased levels of Mdm2-bound Mdm4 in *Rps27l*-null MEFs were due to low Mdm4 levels or due to altered binding affinity or both. We therefore determined the binding affinity of Mdm2 and Mdm4 by two-reciprocal immunoprecipitation (IP) assay in *Rps27l*-wt vs. *Rps27l*-null MEFs under unstressed native condition in the absence of proteasome inhibitor MG132. Although the total cellular level of Mdm4 was much lower in *Rps27l*^*-/-*^ than in *Rps27l*^*+/+*^ MEFs with a ratio of 0.12 vs. 1, the level of Mdm2-bound Mdm4 (Mdm2 IP) was rather similar with a ratio of 0.75 vs. 1. Reciprocally (Mdm4 IP), the level of Mdm4-bound Mdm2 was also higher with a ratio of 0.32 vs. 1. Thus, most Mdm4 in *Rps27l*^-/-^ MEFs was found in the complex with Mdm2. These results clearly demonstrated that in the absence of Rps27l, Mdm2 actually has a higher binding affinity towards Mdm4, which might facilitate Mdm2-mediated Mdm4 degradation, leading to a decreased level of Mdm4. Thus, reduced level of Mdm2-bound Mdm4 in *Rps27l*-null MEFs is mainly due to reduced total level of Mdm4, not due to reduced binding affinity. This newly generated data is now shown in Figure 4 and described in the text.

*5) While the authors demonstrate that Rps27L is physiologically relevant in p53 regulation, defining Rps27L as an inhibitor of p53 simply because its depletion induces p53 may not be valid. As the authors note, the depletion of Rps27L does appear to cause ribosomal stress, and ribosomal stress causes p53 activation, but that does not directly prove that its biological function is to inhibit p53 activation. Does the overexpression of Rps27L in MEFs suppress WT levels of p53? Does Rps27L normally prohibit other ribosomal proteins from stabilizing Mdm2? A logical conclusion from this study is that in the presence of normal levels of Rps27L there is no cellular stress (defective ribosome biogenesis or protein synthesis?) that triggers the p53 response*.

We agree with the reviewers and have replaced this statement of “Rps27l is a physiological inhibitor of p53” by “Rps27l is a physiological regulator of p53”. On the other hand, our data also suggest that Rps27l stabilize Mdm2/Mdm4 complex for targeted p53 degradation, since underlying mechanism by which *Rps27l* deletion triggers ribosomal stress to induce p53 is through Mdm2-mediated Mdm4 degradation to reduce its ligase activity towards p53. In this regard, Rps27l acts as an indirect inhibitor of p53 through optimizing Mdm2/Mdm4 ligase activity towards p53.

[Editors' note: further clarifications were requested prior to acceptance, as described below.]

*1) The authors argue that Rps27l depletion doesn't affect rRNA processing. However, the results shown in Figure S5 C demonstrate that the amount of labeled 18S rRNA was reduced in Rps27l*^*-/-*^
*cells. It will be necessary to quantify these results. The original Conclusion apparently needs to be modified*.

Per the reviewer’s suggestion, we have quantified the amount of labeled 18S rRNA in comparison with that of labeled 47S rRNA, shown in Figure S5C. The results, from two independent isolates of primary MEFs, revealed that the 18S/47S ratio was indeed moderately decreased upon *Rps27l* deletion, although the tendency was similar to the wild type cells. Based upon this quantified measure, we have modified our conclusion by stating now that “p53 induction upon *Rps27l* disruption is unlikely due to aberrant protein synthesis, but reduced rRNA processing may contribute to some extent”. This new quantified data is now shown in Figure 5—figure supplement 1.

*2) It will be necessary to increase the size of the immunofluorescence images in*
Figure 5
*to better appreciate the redistribution of B23 protein form the nucleolus to the nucleus*.

Per the reviewer’s request, we have now increased the size of the images in Figure 5 for better visualization.

*3) The results shown in*
Figure 5
*clearly show that Rpl11 is required for p53 activation upon Rps27l deficiency. It would be important to include Rpl11 in the model (*Figure 9*)*.

Per reviewer’s suggestion, we have included Rpl11 in the model in Figure 9 and discussed it in the text (p53 activation in an Rpl11 dependent manner).

*4) Based on available literature it is completely unexpected that depletion of Rpl5 in wt cells upregulates the p53 protein levels (*Figure 5*). It will be necessary to validate this result with multiple effective siRNAs against Rpl5*.

The reviewer made a valid comment that our original data unexpectedly showed that p53 protein level was increased in wt MEF cells upon silencing of Rpl5. Per reviewer’s request, we have now validated this result using two additional independent Rpl5-targeting siRNAs in yet two independent isolates of primary *Rps27l*^*+/+*^ MEFs from different breeders. The induction of p53 was reproducibly shown (Figure 10). The #3 Rpl5-targeting shRNA is the same one used in Figure 5. We have described this reproducible result in the text.Author response image 1.

5) The authors argue that in addition to RPL11, the binding of Rpl23, Rps7, Rps14 and Rps27 to Mdm2 could be relevant for p53 activation in Rps27l-deficient MEFs. What is the effect of silencing of the above-mentioned RPs on p53 activation in these cells?

We have confirmed that activation of p53 by *Rps27l*-deficiency is dependent on Rpl11, but not Rpl5. Furthermore, we made a novel observation that *Rps27l* disruption caused an increased binding of Mdm2 selectively to Rpl11, Rpl23, Rps7, Rps14, and Rps27, but not to Rpl5, Rps27a, and Rps19. Whether these RPs with enhanced Mdm2 binding indeed contribute to p53 activation in *Rps27l-deficient* MEFs is an interesting topic warranting future investigation. We have now added this statement in the Discussion.

*6) The authors demonstrated in the revised manuscript that p53 levels are higher in Rps27l*^*-/-*^*;p53*^*+/-*^
*than in Rps27l*^*+/+*^*;p53*^*+/-*^
*MEFs (*Figure 8*), arguing that Rps27l deficiency can cause genomic instability independently of p53 loss (*Figure 8*. Is the passage number of MEFs used in experiments shown in*
Figure 8
*the same. This has to be clearly indicated in the legend text*.

We apologize for this omission. The passage numbers of MEFs used in the experiments shown in Figure 8 are P2 and P3. We have now added these passage numbers of MEFs in the legend.